# OPTIMIZING LOSS LANDSCAPE CONNECTIVITY VIA NEURON ALIGNMENT

## ABSTRACT

The loss landscapes of deep neural networks are poorly understood due to their high nonconvexity. Empirically, the local minima of these loss functions can be connected by a learned curve in model space, along which the loss remains nearly constant. Yet, current path finding algorithms do not consider the influence of symmetry in the loss surface caused by weight permutations of the networks corresponding to the minima. We propose a framework to investigate the effect of symmetry on landscape connectivity by directly optimizing the weight permutations of the networks being connected. To learn a locally optimal permutation, we introduce a proximal alternating minimization scheme and prove some convergence guarantees. Using an neuron alignment technique, we derive an inexpensive heuristic for approximating a locally optimal weight permutation. Empirically, optimizing the permutation is critical for efficiently learning a simple, planar, low-loss curve between networks that successfully generalizes. As an application, we also show that there is a performance gain, particularly for models with less parameters, in the accuracy of ensembles constructed from the learned curve with neuron alignment permutation.

## 1 INTRODUCTION

Loss surfaces of neural networks have been of recent interest in the deep learning community both from a numerical (Dauphin et al. (2014); Sagun et al. (2014)) and a theoretical (Choromanska et al. (2014); Safran & Shamir (2015)) perspective. Their optimization yields interesting examples of a high-dimensional non-convex problem, where counter-intuitively gradient descent methods successfully converge to non-spurious minima. Practically, recent advancements in several applications have used insights on loss surfaces to justify their approaches. For instance, Moosavi-Dezfooli et al. (2019) investigates regularizing the curvature of the loss surface to increase the adversarial robustness of trained models.

One interesting question about these non-convex loss surfaces is to what extent trained models, which correspond to local minima, are connected. Here, *connection* denotes the existence of a path between the models, parameterized by their weights, along which loss is nearly constant. There has been conjecture that such models are connected asymptotically, with respect to the width of hidden layers. Recently, Freeman & Bruna (2016) proved this for rectified networks with one hidden layer.

When considering the connection between two neural networks, it is important for us to consider what properties of the neural networks are intrinsic. There is a permutation ambiguity in the indexing of units in a given hidden layer of a neural network, and as a result, this ambiguity extends to the network weights themselves. Thus, there are numerous equivalent points in model space that correspond to a given neural network. This creates weight symmetry in the loss landscape. It is possible that the minimal loss paths between a network and all networks equivalent to a second network could be quite different. If we do not consider the best path among this set, we could fail to see to what extent models are intrinsically connected. Therefore, in this work we propose to develop a technique for more robust model interpolation / optimal connection finding by investigating the effect of weight symmetry in the loss landscape.The analyses and results will give us insight into the geometry of level sets of the loss surfaces of deep networks that are often hard to study theoretically.

**Related Work**   Freeman & Bruna (2016) is one of the first studies to rigorously prove that one hidden layer rectified networks are asymptotically connected and established relevant bounds. Several recent numerical works have shown that parameterized curves along which loss is nearly constant can be successfully learned. Concurrently, Garipov et al. (2018) proposed learning Bezier curves and polygonal chains and Draxler et al. (2018) proposed learning a curve using nudged elastic band energy between two models. Gotmare et al. (2018) showed that these algorithms work even for models trained using different hyperparameters, excluding network architecture. Recently, Kuditipudi et al. (2019) analyzed the connectivity between $\epsilon$-dropout stable networks. This body of work can be seen as the extension of the linear averaging of models studied in (Goodfellow et al., 2014). In fact, concurrent with our work, Singh & Jaggi (2019) applied neuron alignment to model averaging.

The symmetry groups in neural network weight space have long been formally studied (Chen et al., 1993). While permutation ambiguity in the weights has been acknowledged, ostensibly ambiguity due to scaling in the weights has received more attention in research. Numerous regularization approaches based on weight scaling such as in (Cho & Lee, 2017) have been proposed to improve the performance of learned models. Recently, Brea et al. (2019) studied the existence of *permutation plateaus* in which the neurons in the layer of a network can all be permuted at the same cost.

A second line of work studies network similarity. Kornblith et al. (2019) gives a comprehensive review on the topic while introducing centered kernel alignment (CKA) for comparing the behavior of different neural networks. CKA is an improvement over the canonical correlation analysis (CCA) technique introduced in (Raghu et al., 2017) and explored further in (Morcos et al., 2018). Another contribution in this direction is the neuron alignment algorithm from (Li et al., 2016), which showed empirically that two networks of the same architecture learn a subset of similar features.

**Contributions**   We summarize the main contributions of this work as follows:
1. We formalize this problem, and apply a proximal alternating minimization (PAM) scheme to split the problem into iteratively optimizing the permutation of the second model weights and optimizing the curve parameters. We prove convergence of this scheme to a local critical point for feed-forward neural networks which are piece-wise analytic functions and continuously differentiable.
2. Motivated by the neuron alignment technique of (Li et al., 2016) and our PAM framework, we introduce a heuristic for approximating the optimal weight permutation in order to learn *aligned* curves connecting networks up to a symmetry in their weights.
3. We perform experiments on 3 datasets and 4 architectures affirming that more optimal curves can be learned faster with neuron alignment. We observe that this aligned permutation is close to a locally optimal permutation that PAM converges to under the same initialization.
4. For simple networks, we observe a notable improvement in ensemble accuracy when constructing them by sampling the aligned as opposed to the unaligned curve or a set of independent models.

For the structure of this paper, we first review pertinent background on curve finding and neuron alignment in Section 2. Then, we introduce our proposed optimization models and algorithms for curve finding up to a weight permutation in Section 3. In Section 4, we discuss our experiments in detail. In Section 5, we explore the effect of alignment on the performance of model ensembles constructed through sampling along the curve.

## 2   BACKGROUND ON CONNECTIVITY AND ALIGNMENT

In this section we review the existing approaches for loss optima connectivity and neuron alignment.

**Loss Optima Connectivity**   To learn the minimal loss path connecting two $N$-dimensional neural networks, $\boldsymbol{\theta}_1$ and $\boldsymbol{\theta}_2$, we utilize the curve finding approach introduced in (Garipov et al., 2018). Here we search for the path, $\boldsymbol{r} : [0, 1] \mapsto \mathbb{R}^N$, that connects the two models while minimizing the average of the loss function, $\mathcal{L}$, along the path. This problem is formalized in equation 1.

$$\boldsymbol{r}^* = \arg\min_{\boldsymbol{r}} \frac{\int_{t\in[0,1]} \mathcal{L}(\boldsymbol{r}(t))\|\boldsymbol{r}'(t)\|dt}{\int_{t\in[0,1]} \|\boldsymbol{r}'(t)\|dt} \qquad \text{subject to} \quad \boldsymbol{r}(0) = \boldsymbol{\theta}_1, \boldsymbol{r}(1) = \boldsymbol{\theta}_2. \tag{1}$$

For tractability, $\boldsymbol{r}^*$ can be approximated by a parameterized curve $\boldsymbol{r}_\phi$, where $\phi$ denotes the curve parameters. For instance, as described in Section 4, this paper will be using the quadratic Bezier

curve. Computationally, an arclength parameterization, that is $||r'(t)|| = 1$ for all $t$, is assumed to make optimization more computationally feasible. Note that if the endpoint networks are global minima and a flat loss path does exist, then the optimal objective of equation 1 is unchanged.

An equivalent view under the arclength parameterization is that we are minimizing $\mathbb{E}_{t \sim U}[\mathcal{L}(\boldsymbol{r}_\phi(t))]$, where $U$ is the uniform distribution on the unit interval. This view is taken in Algorithm 2 in Appendix C. For clarity, we emphasize that $r_\phi$ denotes the curve on the loss surface between two networks while $\boldsymbol{r}_\phi(t)$ denotes a point on that curve which is a neural network.

**Neuron Alignment**    We give an overview of the neuron alignment framework in (Li et al., 2016). Given input d drawn from the input data distribution $D$, let $\boldsymbol{X}_{l,i,:}^{(1)}(\mathrm{d}) \in \mathbb{R}^k$ represent the activation values of channel $i$ in layer $l$ of network $\boldsymbol{\theta}_1$, where $k$ is the number of units in the channel. As an example, a channel could correspond to one unit in a hidden state or one filter output by a convolutional layer, where $k$ would be 1 or the number of pixels in the filter respectively.

Given networks, $\boldsymbol{\theta}_1$ and $\boldsymbol{\theta}_2$, we define the channel-wise mean and standard deviation for $\theta_1$ in equation 2. We also define the cross-correlation matrix, $C_l^{(1,2)}$, denoting the cross-correlation between each channel in $\boldsymbol{\theta}_1$ and $\boldsymbol{\theta}_2$ in layer $l$.

$$\mu_{l,i}^{(1)} = \mathbb{E}_{\mathrm{d} \sim D}\left[\frac{1}{k}\sum_{a=1}^{k} X_{l,i,a}^{(1)}(\mathrm{d})\right] \qquad \sigma_{l,i}^{(1)} = \sqrt{\mathbb{E}_{\mathrm{d} \sim D}\left[\frac{1}{k}\sum_{a=1}^{k}\left(X_{l,i,a}^{(1)}(\mathrm{d}) - \mu_{l,i}^{(1)}\right)^2\right]}$$

$$C_{l,i,j}^{(1,2)} = \frac{\mathbb{E}_{\mathrm{d} \sim D}\left[\sum_{a=1}^{k}\frac{1}{k}\left(X_{l,i,a}^{(1)}(\mathrm{d}) - \mu_{l,i}^{(1)}\right)\left(X_{l,j,a}^{(2)}(\mathrm{d}) - \mu_{l,j}^{(2)}\right)\right]}{\sigma_{l,i}^{(1)}\sigma_{l,j}^{(2)}} \tag{2}$$

To align the activations in layer $l$ between networks $\boldsymbol{\theta}_1$ and $\boldsymbol{\theta}_2$, the neuron alignment algorithm maximizes the sum of cross-correlation between aligned activations. Equivalently, this finds the permutation, $\boldsymbol{P}_l$, that maximizes the trace of $\boldsymbol{P}_l^T C_{l,:,:}^{(1,2)}$, which is an instance of the linear assignment problem. We formalize this optimization model in equation 3 below, where $K_l$ represents the index set of activations in layer $l$. We recommend Burkard & Cela (1999) as a reference for those unfamiliar with the assignment problem.

$$\max_{\boldsymbol{P}_l} \qquad \mathrm{trace}(\boldsymbol{P}_l^T C_{l,:,:}^{(1,2)}) \tag{3}$$

$$\text{subject to} \qquad \boldsymbol{P}_l 1 = 1, \boldsymbol{P}_l^T 1 = 1, \quad \boldsymbol{P}_l \in \mathbb{Z}_+^{|K_l| \times |K_l|}$$

The alignment technique is visualized in Figure 1a. This displays the cross-correlation matrix for the TinyTen network and CIFAR100 dataset that we discuss later in Section 4. It is clear that the values along the diagonal are much stronger after alignment. Figure 1b displays the mean cross-correlation at each layer between corresponding neurons. This figure also shows the standard deviation of this signal over a set of 3 network pairs. With this *correlation signature* being consistent over different pairs and being increased highly with alignment, we can feel confident that some subset of highly correlated features are being matched.

## 3    OPTIMA CONNECTIVITY CONSIDERING WEIGHT SYMMETRY

We clarify the idea of weight symmetry in a neural network. $\boldsymbol{\theta}_1$ is a neural network on the loss surface parameterized by its weights. A permutation $\boldsymbol{P}_l$ is in $\Pi_{|K_l|}$, the set of permutations on $K_l$, the index set of channels in layer $l$. For simplicity suppose we have an $L$ layer feed-forward network with activation function $\sigma$, weights $\{W_l\}_{l=1}^L$, and input $X_0$. Then the weight permutation ambiguity becomes clear when we introduce the following set of permutations to the feedforward equation:

$$\boldsymbol{Y} := \boldsymbol{W}_L \boldsymbol{P}_{L-1}^T \circ \sigma \circ \boldsymbol{P}_{L-1} \boldsymbol{W}_{L-1} \boldsymbol{P}_{L-2}^T \circ \sigma \circ \boldsymbol{P}_{L-2} \boldsymbol{W}_{L-2} \boldsymbol{P}_{L-3}^T \circ ... \circ \sigma \circ \boldsymbol{P}_1 \boldsymbol{W}_1 \boldsymbol{X}_0 \tag{4}$$

Then we can define the network weight permutation $\boldsymbol{P}$ as the block diagonal matrix, blockdiag$(\boldsymbol{P}_1, \boldsymbol{P}_2, ..., \boldsymbol{P}_{L-1})$. Additionally, $\boldsymbol{P\theta}$ denotes the network parameterized by the weights $[\boldsymbol{P}_1 \boldsymbol{W}_1, \boldsymbol{P}_2 \boldsymbol{W}_2 \boldsymbol{P}_1^T, ..., \boldsymbol{W}_L \boldsymbol{P}_{L-1}^T]$. Note that we omit permutations $\boldsymbol{P}_0$ and $\boldsymbol{P}_L$, as the input and output channels of neural networks have a fixed ordering, so they correspond to the identity $\boldsymbol{I}$. Without much difficulty this framework generalizes for more complicated architectures. We discuss this for residual networks in Appendix E.

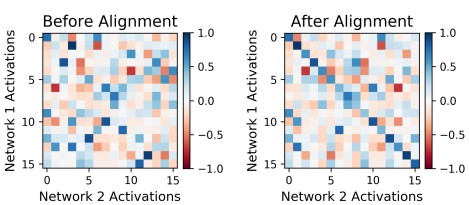
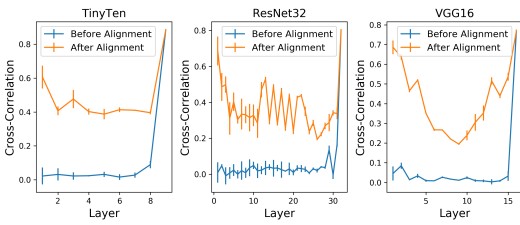

| (a) Cross-correlation between neurons | (b) Mean cross-correlation at each layer |

Figure 1: **(1a)** Cross-correlation between the activations in the first layer of a TinyTen model for CIFAR100. The plot on the left uses the original indices of the second network, while the plot on the right uses the reindexing of the second model consistent with alignment to the first. Note the diagonal of the second matrix is much more positive than the first matrix, which implies a meaningful correspondence between aligned units. **(1b)** The mean cross-correlation between corresponding units for each layer before and after alignment. The standard deviation of this correlation signature over a set of different network pairs is displayed. This shows that the quality of the correspondence between the average pair of units at each layer can be strongly improved through alignment.

### 3.1 CURVE FINDING UP TO SYMMETRY

From equation 4, it becomes clear that the networks $\boldsymbol{\theta}_1$ and $\boldsymbol{P}\boldsymbol{\theta}_1$ share the same structure and intermediate outputs up to indexing. Taking weight symmetry into account, we can find the optimal curve connecting two networks up to symmetry with the model in equation 5.

$$\phi^*, \boldsymbol{P}^* = \arg\min_{\phi,\boldsymbol{P}} \quad \mathbb{E}_{t \sim U}[\mathcal{L}(\boldsymbol{r}_\phi(t))] \tag{5}$$

$$\text{subject to} \quad \boldsymbol{r}_\phi(0) = \boldsymbol{\theta}_1, \boldsymbol{r}_\phi(1) = \boldsymbol{P}\boldsymbol{\theta}_2, \quad \boldsymbol{P} = \text{blockdiag}(\boldsymbol{P}_1, \boldsymbol{P}_2, ..., \boldsymbol{P}_{L-1})$$

$$\boldsymbol{P}_l \in \Pi_{|K_l|} \text{ for } l \in \{1, 2, ..., L-1\}$$

#### 3.1.1 PROXIMAL ALTERNATING MINIMIZATION AS A FRAMEWORK

We first introduce a framework to solve the generalized problem in equation 5. Theoretically, this problem is fairly complicated and hard to analyze. Numerically, approaching the problem directly with first order methods could be computationally intensive as we need to store gradients of $\phi$ and $\boldsymbol{P}$ simultaneously. The problem can be more easily addressed using the method of proximal alternating minimization (PAM) (Attouch et al., 2010). The PAM scheme involves iteratively solving the two subproblems in equation 6. Here we let $Q(\phi, \boldsymbol{P})$ denote the objective function in equation 5. We only consider parameterized forms of $\boldsymbol{r}$ that satisfy the endpoint constraints for all $\phi$ and $\boldsymbol{P}$. For generality, we let $\mathcal{R}$ denote a regularization term on $\phi$.

$$\begin{cases} \boldsymbol{P}^{k+1} = \arg\min_{\boldsymbol{P}} & Q(\phi^k, \boldsymbol{P}) + \frac{1}{2\nu_P}||\boldsymbol{P} - \boldsymbol{P}^k||_2^2 \\ \text{such that} & \boldsymbol{P}_l \in \Pi_{|K_l|} \text{ for } l \in \{1, 2, ..., L-1\} \\ & \boldsymbol{P} = \text{blockdiag}(\boldsymbol{P}_1, \boldsymbol{P}_2, ..., \boldsymbol{P}_{L-1}) \\ \phi^{k+1} = \arg\min_{\phi} & Q(\phi, \boldsymbol{P}^{k+1}) + \mathcal{R}(\phi) + \frac{1}{2\nu_\phi}||\phi - \phi^k||_2^2 \end{cases} \tag{6}$$

Computing the unaligned curve is equivalent to solving the PAM scheme with a very large value of $\nu_P$. In fact, we are able to prove local convergence results for a certain class of networks.

**Theorem 3.1** (Convergence). *Let $\{\phi^{k+1}, \boldsymbol{P}^{k+1}\}$ be the sequence produced by equation 6. Assume that $\boldsymbol{r}_\phi(t)$ corresponds to a feed-forward neural network with activation function $\sigma$ for $t \in [0, 1]$. Assume that $\mathcal{L}$, $r_\phi$, and $\sigma$ are all piece-wise analytic functions in $C^1$ and locally Lipschitz differentiable in $\phi$ and $\boldsymbol{P}$. Additionally, assume $\mathcal{R}$ is piece-wise analytic in the primal variables and bounded below. Lastly, assume that the input data is bounded and the norm of the network weights are constrained to be bounded above. Then the following statements hold:*

1. $Q(\phi^{k+1}, \boldsymbol{P}^{k+1}) + \mathcal{R}(\phi^{k+1}) + \frac{1}{2\nu_\phi}||\phi^{k+1} - \phi^k||_2^2 + \frac{1}{2\nu_P}||\boldsymbol{P}^{k+1} - \boldsymbol{P}^k||_2^2 \le Q(\phi^k, \boldsymbol{P}^k) + \mathcal{R}(\phi^k), \forall k \ge 0$

2. $\{\phi^k, \boldsymbol{P}^k\}$ converges to a critical point of $Q(\phi, \boldsymbol{P}) + \mathcal{R}(\phi)$

*Proof*    See Appendix D

**Remark**    Theorem 3.1 does not extend to neural networks with ReLU activation functions. In Appendix D, we address a technique utilizing this theorem for learning a curve connecting rectified networks while still generating a sequence of iterates with monotonic decreasing objective value.

### 3.1.2    NEURON ALIGNMENT AS AN INITIALIZATION

In spite of convergence guarantees, PAM still requires a good initialization as the loss landscape is nonconvex. This is critical for avoiding convergence to non-global optima. Conceptually, neuron alignment introduced in (Li et al., 2016) is able to match subsets of similar feature representations. Thus, we believe that the permutation on the network weights induced by neuron alignment could be meaningful enough to provide a good initialization of $\boldsymbol{P}$.

In practice, we solve the linear sum assignment problem formulated in equation 3 using the Hungarian algorithm. See Kuhn (1955) for further reading on the Hungarian algorithm. Algorithm 1 summarizes the process for efficiently computing a permutation of the network weights from neuron alignment. For an $L$ layer network with a maximum layer width of $M$, we compute $\boldsymbol{P}$ using a subset of the training data. Then the cost of computing the cross-correlation matrices for all layers is dominated by the forward propogation through the network to accumulate the activations. The running time needed to compute all needed linear assignments is $\mathcal{O}(LM^3)$. This is on the order of the running time associated with one iteration of forward propagation. Then neuron alignment is relatively cheap as the time complexity of computing curves using neuron alignment is on the same order as traditional curve finding. We refer to these different curves as aligned and unaligned.

---

**Data:** Trained Neural Networks $\boldsymbol{\theta}_1$ and $\boldsymbol{\theta}_2$, Subset of Training Data $\mathbf{X}_0$
**Result:** Aligned Neural Networks $\boldsymbol{\theta}_1$ and $\boldsymbol{P}\boldsymbol{\theta}_2$
Initialize $\boldsymbol{P}\boldsymbol{\theta}_2 := [\hat{\boldsymbol{W}}_1^2, \hat{\boldsymbol{W}}_2^2, ..., \hat{\boldsymbol{W}}_L^2]$ as $[\boldsymbol{W}_1^2, \boldsymbol{W}_2^2, ..., \boldsymbol{W}_L^2]$;
**for** *each layer $l$ in $\{1, 2, ..., L-1\}$* **do**
    **for** *each network $j$ in $\{1, 2\}$* **do**
        compute activations, $\mathbf{X}_l^{(j)} = \sigma \circ \boldsymbol{W}_l^j \mathbf{X}_{l-1}^{(j)}$ ;
        for each element in the batch, vectorize $\mathbf{X}_l^{(j)}$ if applicable ;
        compute, $\boldsymbol{Z}_l^{(j)}$, the Z-score normalization of the activations ;
    **end**
    compute the cross-correlation matrix, $\boldsymbol{C}_l^{(1,2)} = \boldsymbol{Z}_l^{(1)} \boldsymbol{Z}_l^{(2)T}$ ;
    compute $\boldsymbol{P}_l$ by solving the assignment problem associated with $\boldsymbol{C}_l^{(1,2)}$ using the Hungarian algorithm ;
    update $\hat{\boldsymbol{W}}_l^2 \to \boldsymbol{P}_l \hat{\boldsymbol{W}}_l^2, \quad \hat{\boldsymbol{W}}_{l+1}^2 \to \hat{\boldsymbol{W}}_{l+1}^2 \boldsymbol{P}_l^T$
**end**

**Algorithm 1:** Computing Permutation via Neuron Alignment

---

## 4    EXPERIMENTS

**Datasets**    In our experiments, we trained neural networks to classify images from CIFAR10 and CIFAR100 (Krizhevsky et al., 2009), as well as STL10 (Coates et al., 2011). The loss function is the cross entropy loss on the softmax of the logits output by the networks. $20\%$ of the images in the training set are used for computing alignments between pairs of models. We augment the data using color normalization, random horizontal flips, random rotation, and random cropping to prevent the models from overfitting on the training set.

**Architectures**    Four different model architectures are used. Table 1 summarizes relevant properties of these architectures. The first architectures considered were the TinySix and TinyTen models. TinyTen, introduced in (Kornblith et al., 2019), is a narrow 10 layer convolutional neural network that uses batch-normalization, ReLU activations, and global average pooling. TinySix is equivalent

to TinyTen with layers 2, 4, 5, and 7 removed. These are useful models for concept testing and allow us to gain insight to networks that are underparameterized. We also include ResNet32 (He et al., 2016) in our experiments to understand the effect of skip connections on curve finding with alignment. VGG16-BN is the third architecture that we considered in our experiments (Simonyan & Zisserman, 2014). VGG16 has significantly more parameters compared to other models. We chose this set of architectures for its varying properties and because of their prevalence in numerical experiments in related literature.

All models used as curve endpoints are trained using stochastic gradient descent. We set a learning rate of $1E{-}1$ that decays by a factor of $0.5$ every 20 epochs. Weight decay of $5E{-}4$ was used for regularization. Each model was trained for 250 epochs, and all models were seen to converge.

Table 1: Properties of models used in this study

| Model | Number of Parameters | Depth | Accuracy | | |
|---|---|---|---|---|---|
| | | | CIFAR10 | CIFAR100 | STL10 |
| TinySix | 34,596 | 6 | $77.0 \pm 0.3$ | $40.1 \pm 0.3$ | $65.1 \pm 0.1$ |
| TinyTen | 86,778 | 10 | $88.7 \pm 0.2$ | $58.1 \pm 0.5$ | $73.8 \pm 0.3$ |
| ResNet32 | 466,906 | 32 | $92.9 \pm 0.2$ | $67.1 \pm 0.5$ | $76.5 \pm 0.3$ |
| VGG16-BN | 15,253,578 | 16 | $93.1 \pm 0.2$ | $70.9 \pm 0.3$ | $72.5 \pm 1.5$ |

**Quadratic Bezier curves**  All curves are parameterized as quadratic Bezier curves. Bezier curves are popular in computer graphics as they can be defined by their *control points*. In the current study, we refer to endpoint models as $\boldsymbol{\theta}_1$ and $\boldsymbol{\theta}_2$ as well as the control point, $\boldsymbol{\theta}_c$. Then $\boldsymbol{r}$ is defined in equation 7

$$\boldsymbol{r}_\phi(t) = (1-t)^2\boldsymbol{\theta}_1 + 2(1-t)t\boldsymbol{\theta}_c + t^2\boldsymbol{\theta}_2. \tag{7}$$

Then $\boldsymbol{\theta}_c$ is the learnable parameter in $\phi$. Of course one could consider more complicated curve parameterizations. In practice, we find a simple curve to be enough for our experiments, and consider the learning of a planar curve along which loss is nearly constant to be significant in itself.

We explicitly define this curve for use in the PAM algorithm in equation 8. Here the curve has been reparameterized so that the control point is a function of the permutation $\boldsymbol{P}$. $\tilde{\boldsymbol{\theta}}_c$ captures the deviation of the control point from the linear midpoint between $\boldsymbol{\theta}_1$ and $\boldsymbol{P}\boldsymbol{\theta}_2$. For PAM, $\tilde{\boldsymbol{\theta}}_c$ is the learnable curve parameter in $\phi$. It is zero initialized so that the initial curve is a linear interpolation between models as in traditional curve finding. This coupling of the control point with the permutation was seen to be critical for the success of PAM.

$$\boldsymbol{r}(t; \tilde{\boldsymbol{\theta}}_c, \boldsymbol{P}) = (1-t)^2\boldsymbol{\theta}_1 + 2(1-t)t\left(\frac{\boldsymbol{\theta}_1 + \boldsymbol{P}\boldsymbol{\theta}_2}{2} + \tilde{\boldsymbol{\theta}}_c\right) + t^2\boldsymbol{P}\boldsymbol{\theta}_2. \tag{8}$$

### 4.1 TRAINING CURVES

For each architecture, we train 12, 6, and 6 different models using different random initializations for CIFAR10, CIFAR100, and STL10 respectively. Thus we have 6 or 3 independent model pairs for a dataset. We learn four classes of curves:

- Unaligned: Solution to algorithm 2 given networks $\boldsymbol{\theta}_1$ and $\boldsymbol{\theta}_2$
- PAM Unaligned: Solution to equation 6 given networks $\boldsymbol{\theta}_1$ and $\boldsymbol{\theta}_2$ with $\boldsymbol{P}^{(0)} = \boldsymbol{I}$
- PAM Aligned: Solution to equation 6 given networks $\boldsymbol{\theta}_1$ and $\boldsymbol{\theta}_2$ with $\boldsymbol{P}^{(0)} = \boldsymbol{P}_{Al}$
- Aligned: Solution to algorithm 2 given networks $\boldsymbol{\theta}_1$ and $\boldsymbol{P}_{Al}\boldsymbol{\theta}_2$

where $\boldsymbol{P}_{Al}$ denotes the permutation learned by neuron alignment (algorithm 1).

We learn PAM curves for all architectures except VGG16, as its size made this computationally prohibitive. We train two sets of each curve class. One set involves the curves learned when the random seed for curve finding is fixed for all model pairs. The other set consists of the curves learned when the random seed is different for each model pair. We find that the learned curves for

different seeds are similar up to reindexing the endpoints. For Figures 2, 3, and 4, we use the first set of curves so that interesting geometric features on the loss surface are not averaged out. For tables and other figures, we use the second set of curves as they are more general.

Table 2: The average accuracy along the curve with standard deviation is reported for each combination of dataset, network architecture, and curve class. This shows that aligned curves not only outperform the unaligned curves which do not consider the permutation ambiguity, they perform as well as the PAM curves which learn a locally optimal permutation [1]. Note that aligned accuracies are typically as high as the trained model accuracies in Table 1.

| Model | Curve Class | CIFAR10 | CIFAR100 | STL10 |
|---|---|---|---|---|
| TinySix | Unaligned | $73.8 \pm 0.4$ | $38.8 \pm 0.2$ | $64.1 \pm 0.3$ |
| | PAM Unaligned | $75.8 \pm 0.4$ | $41.2 \pm 0.3$ | $64.5 \pm 0.3$ |
| | PAM Aligned | $\mathbf{76.3 \pm 0.3}$ | $41.4 \pm 0.5$ | $\mathbf{65.4 \pm 0.2}$ |
| | Aligned | $\mathbf{76.3 \pm 0.3}$ | $\mathbf{41.6 \pm 0.5}$ | $\mathbf{65.4 \pm 0.2}$ |
| TinyTen | Unaligned | $87.2 \pm 0.2$ | $56.0 \pm 0.2$ | $73.7 \pm 0.4$ |
| | PAM Unaligned | $87.9 \pm 0.2$ | $57.1 \pm 0.1$ | $73.9 \pm 0.3$ |
| | PAM Aligned | $\mathbf{88.6 \pm 0.1}$ | $58.5 \pm 0.2$ | $74.0 \pm 0.3$ |
| | Aligned | $\mathbf{88.6 \pm 0.1}$ | $\mathbf{58.7 \pm 0.2}$ | $\mathbf{74.1 \pm 0.3}$ |
| ResNet32 | Unaligned | $92.4 \pm 0.2$ | $66.5 \pm 0.2$ | $76.6 \pm 0.2$ |
| | PAM Unaligned | $92.5 \pm 0.1$ | $66.8 \pm 0.1$ | $76.7 \pm 0.3$ |
| | PAM Aligned | $92.5 \pm 0.1$ | $66.9 \pm 0.2$ | $\mathbf{76.9 \pm 0.2}$ |
| | Aligned | $\mathbf{92.9 \pm 0.2}$ | $\mathbf{67.7 \pm 0.1}$ | $76.7 \pm 0.2$ |
| VGG16 | Unaligned | $93.0 \pm 0.1$ | $70.7 \pm 0.1$ | $74.5 \pm 1.0$ |
| | Aligned | $\mathbf{93.3 \pm 0.1}$ | $\mathbf{71.6 \pm 0.1}$ | $\mathbf{74.7 \pm 0.8}$ |

### 4.1.1 NEURON ALIGNMENT

First, we investigate the effects of using neuron alignment as a heuristic for curve finding up to symmetry. That is, we are determining some weight permutation $P_{Al}$ and then finding the curve between networks $\theta_1$ and $P_{Al}\theta_2$. The unaligned and aligned curves were both trained for 200 epochs using stochastic gradient descent with an annealing learning rate. The training of these curves share the same hyperparameters as the training of the individual models.

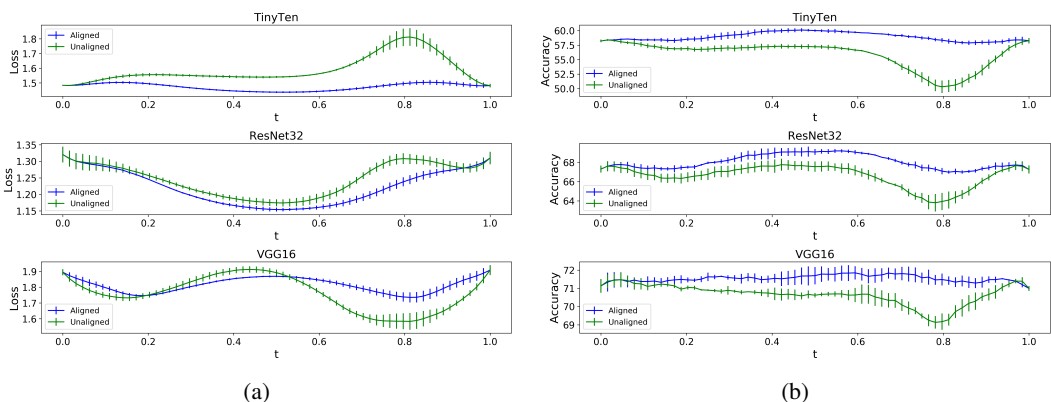

(a)      (b)

Figure 2: Test loss (left) and accuracy (right) of the learned quadratic Bezier curve between model endpoints trained on CIFAR100. Results are compared for aligned (blue) and unaligned (green) curves. This shows that aligned curves have better generalization performance and do not suffer from large drops in accuracy typical for unaligned curves.

The test accuracy can be seen for each dataset and curve class in Table 2. Clearly, the aligned curves outperform the unaligned curve. In many cases, the average accuracy along the aligned curves in

---

[1]Strictly speaking, the algorithm converges to a local optima in the convex relaxation of the domain. The learned permutation is the projection of this optima to the feasible set.

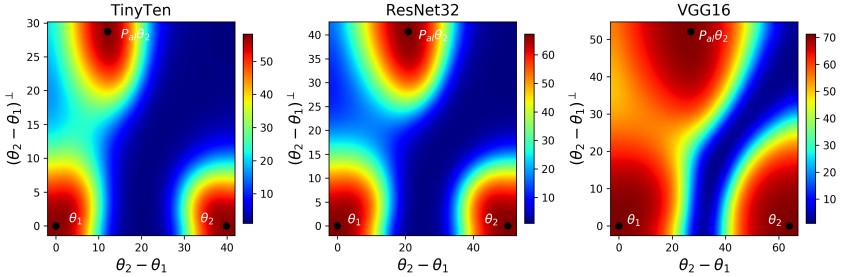

Figure 3: Test accuracy on CIFAR100 across the plane containing $\boldsymbol{\theta}_1$, $\boldsymbol{\theta}_2$, and $\boldsymbol{P}_{al}\boldsymbol{\theta}_2$, where $\boldsymbol{P}_{al}$ is determined using neuron alignment. This plane contains the two different intializations used in our curve finding experiments. The default initialization, $\boldsymbol{\theta}_2 - \boldsymbol{\theta}_1$, and the aligned initialization, $\boldsymbol{P}_{al}\boldsymbol{\theta}_2 - \boldsymbol{\theta}_1$. This shows that the aligned initialization is notably better.

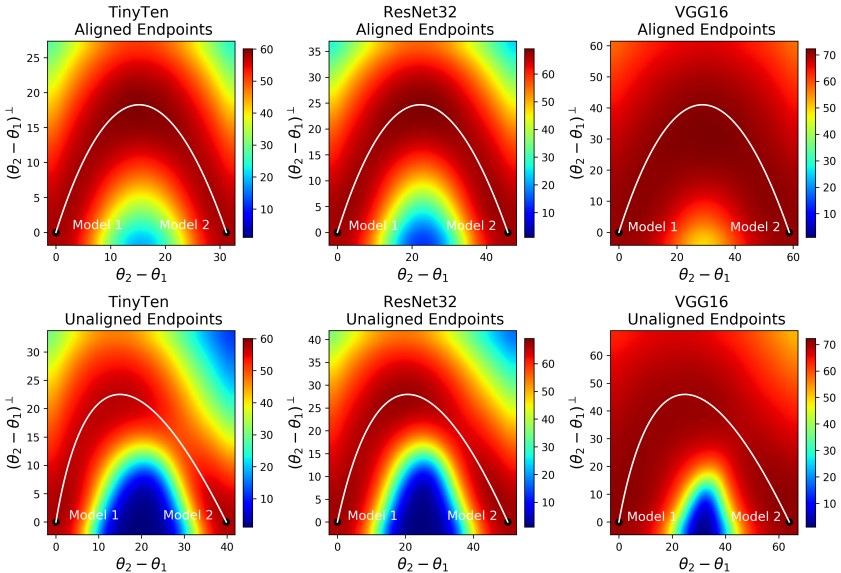

Figure 4: Test accuracy on CIFAR100 across the plane containing the bezier curve, $\boldsymbol{r}_\phi(t)$.

comparable to the trained models used as endpoints. Table 4 contains the minimum accuracy along the curve, indicating that while aligned curves do not suffer from the same generalization gap that unaligned curves are prone to. Finally, Table 5 contains the training loss for each case at convergence. Overall, it is clear that the strongest gains from using alignment are in the case of underparameterized networks. As seen in Table 2, the largest increase in performance is for TinySix on CIFAR100 while the smallest gain is made for STL10 on VGG16. This is inline with observations by (Freeman & Bruna, 2016).

The test loss and accuracy along the learned curves for CIFAR100 are shown in Figure 2. The corresponding Fourier transform of the loss along the curve for assessing curve smoothness is displayed in Figure 11. We observe that, as expected, the accuracy at each point along the aligned curve exceeds that of the unaligned curve, while the loss along the curve is also smoother with neuron alignment. We are comparing loss and accuracy at the curve parameter, $t$. Noteworthy is the prominent presence of the accuracy barrier along the unaligned curve around $t$ at $0.8$ for all models. This accuracy barrier corresponds to a clear loss barrier for Tiny-10 and ResNet32. In contrast, for VGG16 there is a valley in the loss function at this point on the unaligned curve with worse generalization performance. Overall, we find that loss along the aligned curves varies more smoothly as seen in Figure 11, and this leads to better generalization of the interpolated models.

Figure 3 displays the planes which contains the initializations for curve finding. It is clear that the aligned initialization has better objective value. This can also be seen for the other datasets in Figure 7. The planes containing the learned curves are displayed in Figures 4 and 13. These are the planes containing $\boldsymbol{\theta}_1$, $\boldsymbol{P}\boldsymbol{\theta}_2$, and $\boldsymbol{\theta}_c$, although the control point is out of bounds of the figure. The axis is determined by Gram-Schmidt orthonormalization. The loss displayed on the planes containing the linear initializations and the Bezier curves can be seen in Figures 6 and 12.

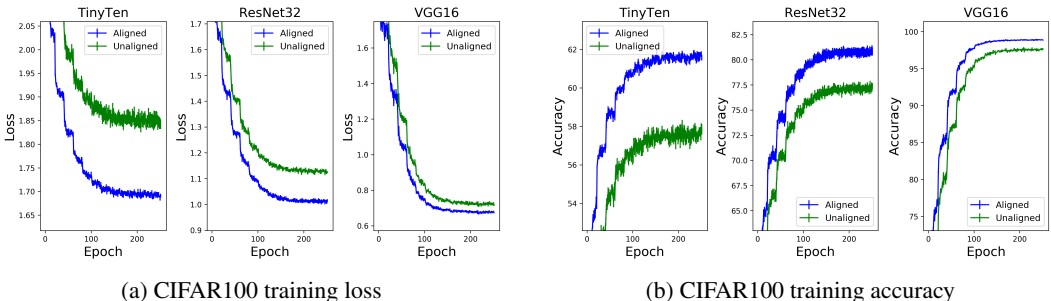

(a) CIFAR100 training loss          (b) CIFAR100 training accuracy

Figure 5: The training loss and training accuracy for learning the quadratic Bezier curve between model endpoints. These are compared for aligned and unaligned curves. This shows that the training of aligned curves converges to lower loss value in less epochs than the training of unaligned curves.

Practically, the neuron alignment heurisitc for determining the permutation $\boldsymbol{P}$ may be enough and avoids performing more complicated optimization. We see this by noting the relative flatness of the accuracy along the aligned curves in Figure 2b. Additionally, Figure 5 indicates much faster convergence when learning $\phi$ using neuron alignment, which is quite impressive. For example, the aligned curve takes $100$ epochs less to achieve the training accuracy that the unaligned curve converges to, when TinyTen is used on CIFAR100. Even for VGG16, the aligned curve reaches the milestone $40$ epochs earlier. Figures 8 and 9 display these training curves for the additional datasets.

While these observations are promising, we intend to provide insight into why neuron alignment works, whih is addressed in Appendix F where we investigate how the alignment is preserved along the learned curve. We find that the midpoints of the unaligned curves are highly aligned to each endpoint even though the endpoints themselves are weakly aligned at best. Then, we find that curve finding is essentially trying to smoothly interpolate similar feature representations. Sensibly, neuron alignment of the endpoints makes this task easier.

### 4.1.2 PROXIMAL ALTERNATING MINIMIZATION

Proximal alternating minimization provides a comprehensive formulation for learning the weight permutation $\boldsymbol{P}$ directly, coupled with some convergence guarantees. We find that curves learned using PAM perform better than the unaligned curves as seen in Table 2. As was the case for the aligned curves, this performance gain is more notable in underparameterized models. Notably, the aligned curves perform comparably to PAM aligned. This indicates that $\boldsymbol{P}_{Al}$ is already close to the locally optimal permutation when $\boldsymbol{P}_{Al}$ is chosen as the initialization for PAM. Additionally, the performance gain of PAM Aligned over PAM Unaligned shows that this permutation is not easy to learn when $\boldsymbol{P}^{(0)}$ is not necessarily close. Then training aligned curves is an inexpensive way to approximate the solution to a rigorous optimization method with good initialization. We stress that this observation shows that the gain from neuron alignment is not trivial.

We now address some technical points in the training of the PAM curves. To learn each curve, we perform 4 iterations of PAM. The permutation subproblem entails 20 epochs of projected stochastic gradient descent to the set of doubly stochastic matrices. This is done as the set of doubly stochastic matrices is the convex relaxation of the set of permutations. This projection is accomplished through 20 iterations of alternating projection of the updated permutation to the set of nonnegative matrices and the set of matrices with row and column sum of $1$. After the 20 epochs of projected gradient descent, each layer permutation is projected to the set of permutations, $\Pi_{|K_l|}$. The curve

parameter subproblem, which optimizes $\tilde{\boldsymbol{\theta}}_c$ from equation 8, entails 40 epochs of stochastic gradient descent. The same hyperparameters are used as in training the endpoint models. The learning rates are annealed with each iteration of PAM. This training can be seen for CIFAR100 in Figure 10.

The outlier case in Table 2 is for ResNet32 where Aligned notably outperforms PAM Aligned. This result is due to the integral nature of the problem. Essentially, during PAM Aligned, the scheme learns a more optimal doubly stochastic matrix that actually projects to a worse permutation at the end of the permutation subproblem. This is a known possible negative, when solving a problem in the convex hull and then projecting back to the feasible set.

## 5 APPLICATION: CONSTRUCTING ENSEMBLES ALONG ALIGNED CURVES

We investigate if the diversity of models along the curve suffers due to alignment through the lens of constructing network ensembles. We consider the simple ensemble that performs classification by averaging the probability distributions output by the individual networks. In our experiment, we look at four cases. The ensemble formed by considering the curve endpoints, the ensembles formed by the curve endpoints and the midpoint, and an ensemble of the curve endpoints and a third independent model. We consider ensembling only on the CIFAR100 dataset with results for the aligned and unaligned case being summarized in Table 3. We see that at best, ensembles constructed by sampling along the unaligned curve perform as well as the independent ensemble. When constructing ensembles by sampling along the aligned curve, those outperform the independent ensemble for the TinySix, TinyTen, and Resnet32 case and show comparable performance for VGG16. The enhanced performance is most obvious on TinyTen. This result makes sense since ensembling has been reported to lead to better performance increase for simpler networks (Ju et al., 2018). This result is encouraging as ensembles of simple models are common in practice. We also note that the performance gain from the aligned ensemble is comparable in magnitude to the gains of Fast Geometric Ensembling in (Garipov et al., 2018) over Snapshot Ensembling (Huang et al., 2017).

Table 3: Accuracy of model ensembles constructed from the curve connecting trained models on the CIFAR100 dataset. Standard deviation is reported as well. All ensembles function by averaging the predictions of their component models. The curve ensembles are constructed by sampling the curve at the values of $t$ listed. The independent ensemble consists of three independently trained models.

| curve parameter $t$ | Curve Ensembles (%) | | | Independent Ensembles (%) |
|---|---|---|---|---|
| | $\{0.0, 1.0\}$ | $\{0.0, 0.5, 1.0\}$ | | |
| | | Unaligned | Aligned | |
| TinySix | $41.80 \pm 0.15$ | $42.66 \pm 0.48$ | $\mathbf{43.82 \pm 0.78}$ | $42.05 \pm 0.26$ |
| TinyTen | $61.23 \pm 0.36$ | $61.39 \pm 0.25$ | $\mathbf{62.40 \pm 0.30}$ | $61.76 \pm 0.01$ |
| ResNet32 | $71.35 \pm 0.30$ | $72.13 \pm 0.12$ | $\mathbf{72.33 \pm 0.12}$ | $72.03 \pm 0.08$ |
| VGG16 | $74.00 \pm 0.17$ | $74.69 \pm 0.22$ | $\mathbf{74.91 \pm 0.10}$ | $74.88 \pm 0.06$ |

## 6 DISCUSSION AND FUTURE WORK

We generalize the curve finding problem by removing the weight symmetry ambiguity associated with the endpoint models. The optimal permutation of these weights can be approximated using neuron alignment. We find empirically that this approximation performs comparably to a proximal alternating scheme with the same initialization which learns a locally optimal permutation. Additionally, we prove that this PAM scheme has some convergence guarantees. Empirically, we show that neuron alignment can be used to successfully and efficiently learn optimal connections between neural nets. Addressing the ambiguity of weight symmetry is critical for learning planar curves on the loss surface along which accuracy is mostly constant. Our results hold true over a range of datasets and network architectures. With neuron alignment, these curves can be trained in less epochs and to higher accuracy. We also see a modest to notable increase in the performance of ensembling.

Future work will include gaining a deeper *theoretical understanding* of how neuron alignment affects curve finding dynamics. We plan to further explore how similar feature representations between models are smoothly interpolated by these learned curves, and how this relates to network training.

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

**Supplementary Material for Optimizing Loss Landscape Connectivity via Neuron Alignment**

# A    ADDITIONAL FIGURES

## A.1    PLANES CONTAINING LINEAR INITIALIZATIONS

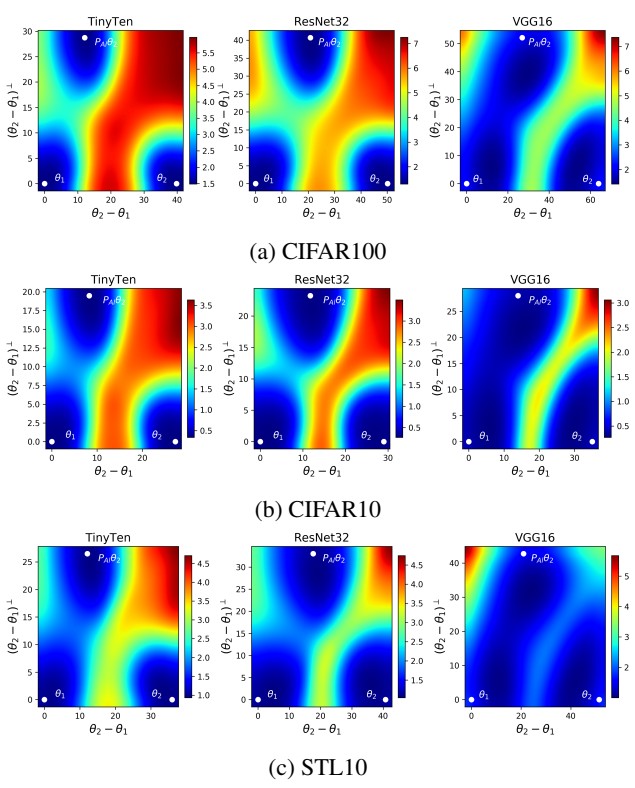

Figure 6: Test loss on plane containing $\boldsymbol{\theta}_1$, $\boldsymbol{\theta}_2$, and $\boldsymbol{P}_{al}\boldsymbol{\theta}_2$.

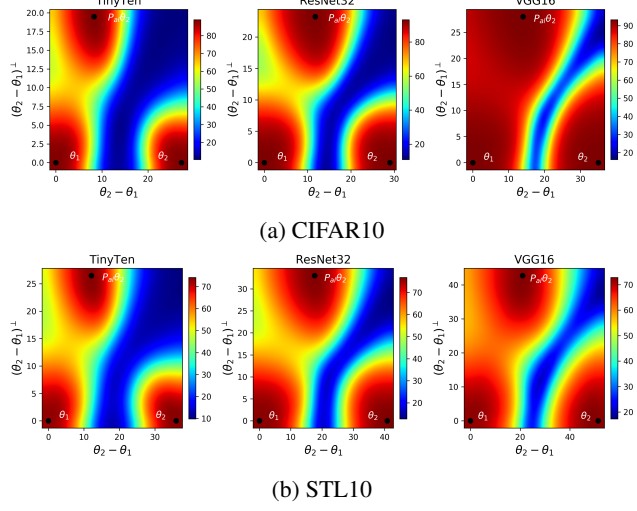

Figure 7: Test accuracy on plane containing $\boldsymbol{\theta}_1$, $\boldsymbol{\theta}_2$, and $\boldsymbol{P}_{al}\boldsymbol{\theta}_2$.

## A.2 TRAINING

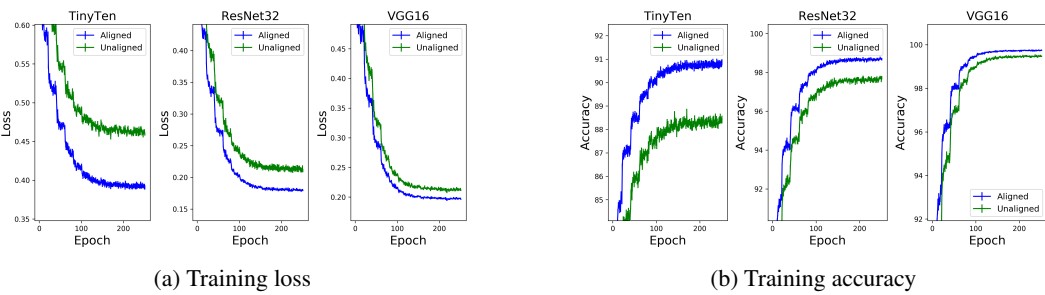

(a) Training loss           (b) Training accuracy

Figure 8: CIFAR10

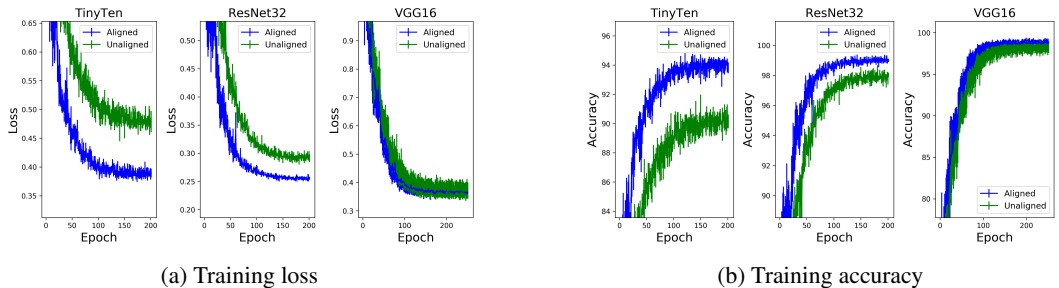

(a) Training loss           (b) Training accuracy

Figure 9: STL10

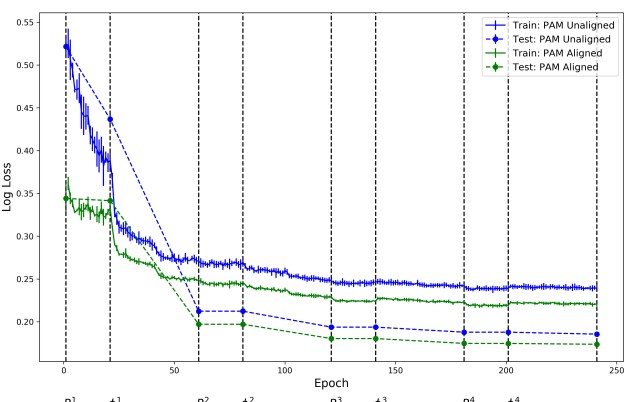

Figure 10: Log loss over a run of the proximal alternating minimization scheme on TinyTen for CIFAR100. The scheme consists of 20 epochs of projected SGD to solve the permutation subproblem, followed by 40 epochs of SGD to solve the curve parameter subproblem. Vertical lines denote the change in different subproblem iterations. This shows that neuron alignment provides a much better intialization for PAM, and this permutation initialization is close to being locally optimal.

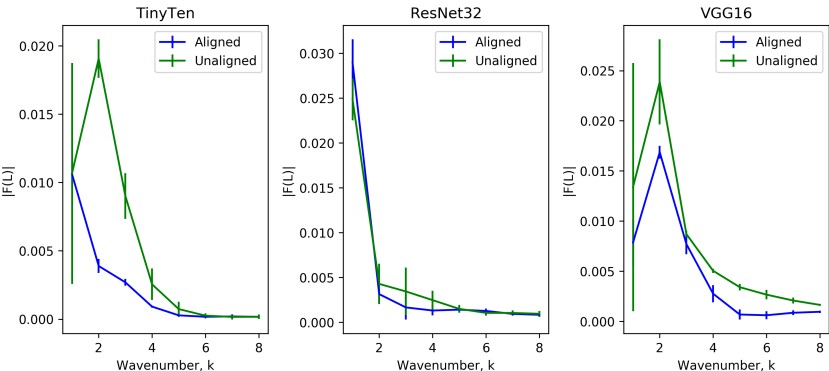

Figure 11: Fourier transform of CIFAR100 loss curve. Notice that the absolute value of the transform is lower for the aligned case at higher modes/wavenumbers. In spectral terms, this shows that the average aligned curve is less oscillatory than the unaligned curve. This is a rigorous way to measure the *smoothness* of a curve.

### A.3 PLANES CONTAINING LEARNED BEZIER CURVES

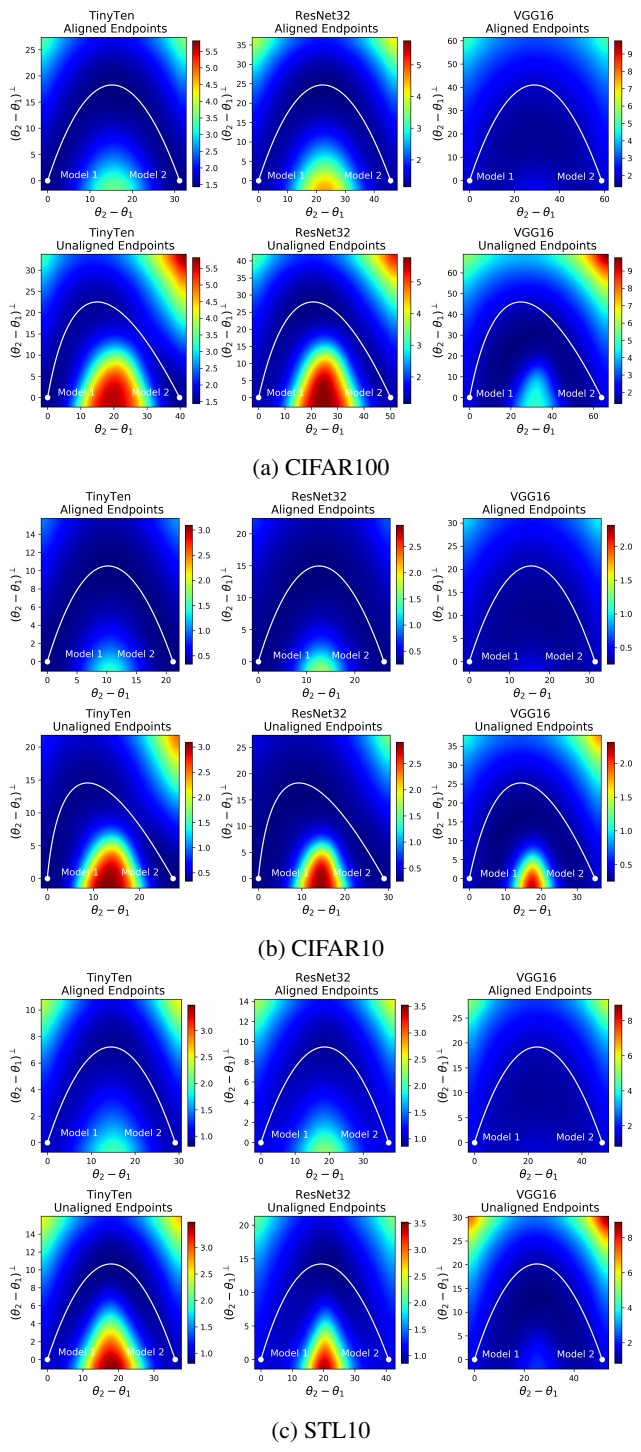

Figure 12: Test loss on plane containing learned curve, $\boldsymbol{r}_\phi(t)$.

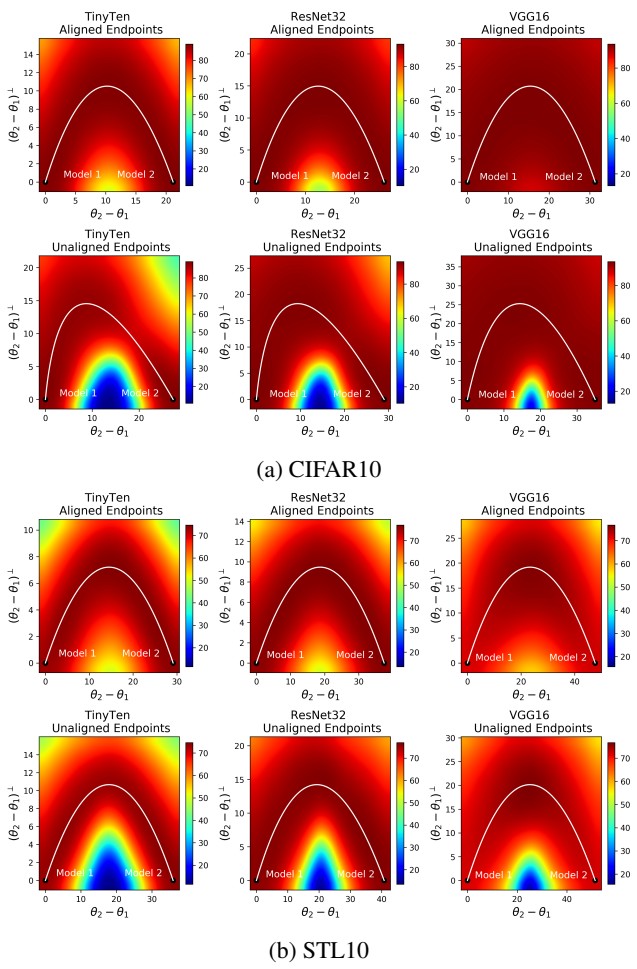

Figure 13: Test accuracy on plane containing learned curve, $\boldsymbol{r}_\phi(t)$.

# B  ADDITIONAL RESULTS

Table 4: The minimum accuracy along the curve with standard deviation is reported for each combination of dataset, network architecture, and curve class.

| Model | Endpoints | CIFAR10 | CIFAR100 | STL10 |
|---|---|---|---|---|
| TinySix | Unaligned | $71.6 \pm 0.6$ | $35.9 \pm 2.3$ | $62.6 \pm 0.6$ |
| | PAM Unaligned | $74.8 \pm 0.5$ | $39.8 \pm 0.2$ | $63.4 \pm 0.4$ |
| | PAM Aligned | $\mathbf{75.7 \pm 0.4}$ | $\mathbf{39.9 \pm 0.3}$ | $\mathbf{64.8 \pm 0.1}$ |
| | Aligned | $75.6 \pm 0.5$ | $\mathbf{39.9 \pm 0.2}$ | $\mathbf{64.8 \pm 0.1}$ |
| TinyTen | Unaligned | $85.0 \pm 1.2$ | $53.2 \pm 1.1$ | $72.3 \pm 0.6$ |
| | PAM Unaligned | $87.2 \pm 0.3$ | $56.1 \pm 0.1$ | $72.6 \pm 0.5$ |
| | PAM Aligned | $88.0 \pm 0.1$ | $\mathbf{57.7 \pm 0.3}$ | $\mathbf{73.5 \pm 0.3}$ |
| | Aligned | $\mathbf{88.6 \pm 0.1}$ | $\mathbf{57.7 \pm 0.4}$ | $73.4 \pm 0.3$ |
| ResNet32 | Unaligned | $91.2 \pm 0.7$ | $64.7 \pm 0.4$ | $75.7 \pm 0.3$ |
| | PAM Unaligned | $91.7 \pm 0.3$ | $65.1 \pm 1.0$ | $76.0 \pm 0.4$ |
| | PAM Aligned | $91.8 \pm 0.2$ | $65.3 \pm 0.6$ | $\mathbf{76.1 \pm 0.3}$ |
| | Aligned | $\mathbf{92.6 \pm 0.2}$ | $\mathbf{66.6 \pm 0.1}$ | $76.0 \pm 0.2$ |
| VGG16 | Unaligned | $92.6 \pm 0.1$ | $69.6 \pm 0.4$ | $\mathbf{71.3 \pm 1.8}$ |
| | Aligned | $\mathbf{93.0 \pm 0.1}$ | $\mathbf{70.8 \pm 0.1}$ | $\mathbf{71.3 \pm 1.8}$ |

Table 5: The training loss with standard deviation is reported for each combination of dataset, network architecture, and curve class.

| Model | Endpoints | CIFAR10 | CIFAR100 | STL10 |
|---|---|---|---|---|
| TinySix | Unaligned | $0.896 \pm 0.008$ | $2.636 \pm 0.008$ | $0.931 \pm 0.014$ |
| | PAM Unaligned | $0.826 \pm 0.006$ | $2.501 \pm 0.015$ | $0.914 \pm 0.010$ |
| | PAM Aligned | $\mathbf{0.813 \pm 0.008}$ | $\mathbf{2.489 \pm 0.018}$ | $\mathbf{0.863 \pm 0.005}$ |
| | Aligned | $0.817 \pm 0.006$ | $2.490 \pm 0.018$ | $0.864 \pm 0.004$ |
| TinyTen | Unaligned | $0.460 \pm 0.004$ | $1.839 \pm 0.010$ | $0.479 \pm 0.018$ |
| | PAM Unaligned | $0.419 \pm 0.005$ | $1.733 \pm 0.013$ | $0.469 \pm 0.000$ |
| | PAM Aligned | $\mathbf{0.387 \pm 0.003}$ | $\mathbf{1.662 \pm 0.007}$ | $0.393 \pm 0.007$ |
| | Aligned | $0.392 \pm 0.003$ | $1.693 \pm 0.008$ | $\mathbf{0.389 \pm 0.007}$ |
| ResNet32 | Unaligned | $0.212 \pm 0.005$ | $1.124 \pm 0.005$ | $0.292 \pm 0.007$ |
| | PAM Unaligned | $0.191 \pm 0.002$ | $1.093 \pm 0.011$ | $0.283 \pm 0.001$ |
| | PAM Aligned | $0.192 \pm 0.002$ | $1.086 \pm 0.011$ | $0.283 \pm 0.001$ |
| | Aligned | $\mathbf{0.180 \pm 0.002}$ | $\mathbf{1.011 \pm 0.002}$ | $\mathbf{0.256 \pm 0.002}$ |
| VGG16 | Unaligned | $0.212 \pm 0.002$ | $0.710 \pm 0.004$ | $0.372 \pm 0.029$ |
| | Aligned | $\mathbf{0.198 \pm 0.001}$ | $\mathbf{0.676 \pm 0.005}$ | $\mathbf{0.366 \pm 0.015}$ |

## C  ALGORITHMS

This section contains algorithms described in Section 2.

**Data:** Two trained models, $\boldsymbol{\theta}_1$ and $\boldsymbol{\theta}_2$
**Result:** A parameterized curve, $\boldsymbol{r}_\phi$, connecting $\boldsymbol{\theta}_1$ and $\boldsymbol{\theta}_2$ along which loss is flat
Initialize $\boldsymbol{r}_\phi(t)$ as $\boldsymbol{\theta}_1 + t(\boldsymbol{\theta}_2 - \boldsymbol{\theta}_1)$;
**while** *not converged* **do**
    **for** *batch in dataset* **do**
        sample point $t_0$ in $[0, 1]$;
        compute loss $L(\boldsymbol{r}_\phi(t_0))$ ;
        optimization step on network $\boldsymbol{r}_\phi(t_0)$ to update $\phi$ ;
    **end**
**end**

**Algorithm 2:** Curve Finding (Garipov et al., 2018)

In this algorithm, the optimization step can correspond to a variety of techniques. In this paper, we use traditional stochastic gradient descent to update the curve parameters $\phi$. Notice that stochasticity is introduced by the sampling of $t$ as well as the training data.

For the purpose of computing validation loss and test loss for $\boldsymbol{r}_\phi$, important care must be given for networks that contain batch normalization layers. This is because batch normalization aggregates running statistics of the network output that are used when evaluating the model. Though, $\boldsymbol{r}_\phi(t_0)$ gives the weights for the model at point $t_0$, the running statistics need to be aggregated for each normalization layer. In practice, this can be done by training the model for one epoch, while freezing all learnable parameters of the model. Since batch statistics would need to be computed for each point sampled along the curve, it happens that computing the validation or test loss of the curve $\boldsymbol{r}_\phi$ is more expensive than an epoch of training.

## D  PROOFS

For the following proofs, we first establish and more rigorously define some terminology. We first discuss an important abuse of notation. For clarity the parameterized curve connecting networks under some permutation $\boldsymbol{P}$ that has been written as $\boldsymbol{r}_\phi(t)$ will now sometimes be referred to as $\boldsymbol{r}(t; \phi, \boldsymbol{P})$.

**Feed-forward neural networks**   In this section, we will be analyzing feed-forward neural networks. We let $\boldsymbol{X}_0 \in \mathbb{R}^{m_0 \times d}$ be the input to the neural network, $d$ samples of dimension $m_0$. Then we let $\boldsymbol{W}_i \in \mathbb{R}^{m_i \times m_{i-1}}$ denote the network weights mapping from layer $l-1$ to layer $l$. Additionally, $\sigma$ denotes the pointwise activation function. Then we can express the output of a feed-forward neural network, $\boldsymbol{Y}$, as:

$$\boldsymbol{Y} := \boldsymbol{W}_L \sigma \circ \boldsymbol{W}_{L-1} \sigma \circ \boldsymbol{W}_{L-2}...\sigma \circ \boldsymbol{W}_1 \boldsymbol{X}_0 \tag{9}$$

To include biases, $\{\boldsymbol{b}_i\}_{i=1}^L$, we simply convert to homogeneous coordinates,

$$\hat{\boldsymbol{X}}_0 = \begin{bmatrix} \boldsymbol{X}_0 \\ 1 \end{bmatrix}, \qquad \hat{\boldsymbol{W}}_i = \begin{bmatrix} \boldsymbol{W}_i & \boldsymbol{b}_i \\ \boldsymbol{0} & 1 \end{bmatrix}, \qquad \hat{\boldsymbol{Y}} = \begin{bmatrix} \boldsymbol{Y} \\ 1 \end{bmatrix} \tag{10}$$

In all proofs, these terms are interchangeable.

**Huberized ReLU**   The commonly used ReLU function is defined as $\sigma(t) := \max(0, t)$. However, this function is not in $C^1$ and hence not locally Lipschitz differentiable. This makes conducting analysis with this function difficult. Thus, we will approach studying it through the lens of the huberized ReLU function, defined as:

$$\sigma_\delta(t) := \begin{cases} 0 & \text{for } t \le 0 \\ \frac{1}{2\delta}t^2 & \text{for } 0 \le t \le \delta \\ t - \frac{\delta}{2} & \text{for } \delta \le t \end{cases} \tag{11}$$

It is clear that $\sigma_\delta$ is a $C^1$ approximation of $\sigma$ such that $||\sigma - \sigma_\delta||_\infty = \frac{\delta}{2}$. Using huberized forms of loss functions for analysis is a fairly common technique such as in (Xu et al., 2016) which studies huberized support vector machines.

**Kurdyka-Lojasiewicz property**   The function $f$ is said to have the Kurdyka-Lojasiewics (KL) property at $\bar{x}$ if there exist $\nu \in (0, +\infty]$, a neighborhood $U$ of $\bar{x}$ and a continuous concave function $\psi : [0, \nu) \to \mathbb{R}_+$ such that:

- $\psi(0) = 0$
- $\psi$ is $C^1$ on $(0, \nu)$
- $\forall s \in (0, \nu), \psi'(s) > 0$
- $\forall x \in U \cap [f(\bar{x}) < f < f(\bar{x}) + \nu]$, the Kurdyka-Lojasiewics inequality holds

$$\psi'(f(x) - f(\bar{x}))\text{dist}(0, \partial f(x)) \ge 1. \tag{12}$$

Here $\partial f$ denotes the subdifferential of $f$. Informally, a function that satisfies this inequality is one whose range can be reparameterized such that a kink occurs at its minimum. More intuitively, if $\psi$ has the form, $s^{1-\theta}$, and $f$ is differentiable on $(0, \nu)$, then the inequality reduces to

$$\frac{1}{(1-\theta)}|f(x)|^\theta \le ||\nabla f(x)|| \tag{13}$$

**Semialgebraic function**   A subset of $\mathbb{R}^n$ is semialgebraic if it can be written as a finite union of sets of the form

$$\{x \in \mathbb{R}^n : p_i(x) = 0, q_i(x) < 0, i = \{1, 2, ..., p\}\}$$

where $p_i$ and $q_i$ are real polynomial functions. A function $f : \mathbb{R}^n \to \mathbb{R} \cup \{+\infty\}$ is said to be semialgebraic if its graph is a semialgebraic subset of $\mathbb{R}^{n+1}$.

**Subanalytic function**   Globally subanalytic sets are sets of the form $\{(x, t) \in [-1, 1]^n \times \mathbb{R} : f(x) = t\}$ where $f : [-1, 1]^n \to \mathbb{R}$ is an analytic function that can be extended analytically on a neighborhood of the interval $[-1, 1]^n$. A function is subanalytic if its graph is a globally subanalytic set.

### D.1 PROOF OF THEOREM 3.1

To prove this, we need that our problem meets the conditions required for local convergence of proximal alternating minimization (PAM) described in (Attouch et al., 2010). This requires the following:

1. Each term in the objective function containing only one primal variable is bounded below and lower semicontinuous.

2. Each term in the objective function which contains both variables is in $C^1$ and is locally Lipschitz differentiable.

3. The objective function satisfies the Kurdyka-Lojasiewicz (KL) property.

First we reformulate the problem so that it becomes unconstrained. Let $\chi$ denote the indicator function, where:

$$\chi_C(t) := \begin{cases} 0, & \text{for } t \in C \\ +\infty, & \text{otherwise} \end{cases} \tag{14}$$

This problem contains two hard constraints. First, each permutation matrix, $P_l$, must clearly be restricted to the set of permutation matrices of size $|K_l|$, $\Pi_{|K_l|}$. Additionally, it is assumed that the norm of the weights are bounded above. Without loss of generality, let $K_W$ denote an upper bound valid for all the weights. We denote the set of weights that satisfy the norm constraint as $\{A : ||A||_2^2 \leq K_W\}$. Then equation 5 with added regularization is equivalent to:

$$\phi^*, \boldsymbol{P}^* = \arg\min_{\phi, \boldsymbol{P}} \quad Q(\phi, \boldsymbol{P}) + \mathcal{R}(\phi) + \sum_{l=1}^{L-1} \chi_{\Pi_{|K_l|}}(\boldsymbol{P}_l) + \sum_{l=1}^{L} \chi_{\{A:||A||_2^2 < K_W\}}(W_l) \tag{15}$$

We now address each requirement for local convergence.

1. By the theorem, $\mathcal{R}$ is assumed to be bounded below and lower semicontinuous. It is easy to see from equation 14 that the sum of indicator functions are bounded below and lower semicontinuous.

2. Now we consider the form of the function, $Q(\phi, \boldsymbol{P})$. It has been defined as

$$\int_{t=0}^{1} \mathcal{L}(\boldsymbol{r}(t; \phi, \boldsymbol{P})) dt$$

We know that $\boldsymbol{r}(t; \phi, \boldsymbol{P})$ corresponds to a feed-forward neural network. Then $Q$ can be expressed as:

$$\int_{t=0}^{1} \mathcal{L}\left(W_L(t; \phi, \boldsymbol{P})\sigma \circ W_{L-1}(t; \phi, \boldsymbol{P})...\sigma \circ W_1(t; \phi, \boldsymbol{P})X_0\right) dt \tag{16}$$

with weight matrices $W_i$ and activation function $\sigma$. It becomes clear that for $Q(\phi, \boldsymbol{P})$ to be in $C^1$ and locally Lipschitz differentiable, the same must be true for $\mathcal{L}$, $\sigma$, and $\{W_i\}_{i=1}^{L}$. The first two are true as they are assumptions of the theorem. Since, $r_\phi$ is in $C^1$ and locally Lipschitz differentiable in the primal variables, then this is also true for all $W_i$. Thus, $Q(\phi, \boldsymbol{P})$ is in $C^1$ and locally Lipschitz differentiable.

3. To satisfy the KL property, the objective function must be a *tame* function (Attouch et al., 2010). Rigorously, this means that the graph of the function belongs to an o-minimal structure, a concept from algebraic geometry. We refer curious readers to (van den Dries & Speissegger, 2002) for further reference.

   First, we note that $Q(\phi, \boldsymbol{P})$ is piece-wise analytic. This is because $Q$ is a composition of piece-wise analytic functions, $\mathcal{L}$, $\sigma$, and $r_\phi$. Additionally, because the input data is bounded and the norm of the weight matrices are bounded, it follows that the domain of $Q$ is bounded. Since, $Q$ is a piece-wise analytic function with bounded domain, it follows that $Q$ is a subanalytic function. The boundedness of the domain is an important detail here. This is because analytic functions are not necessarilly subanalytic unless their domain is bounded; a popular example of such a function is the exponential function.

The regularization function, $\mathcal{R}$, is assumed to be a piece-wise analytic function. It follows from the previous reasoning that $\mathcal{R}$ has bounded domain. Thus, $\mathcal{R}$ is a subanalytic function.

We now consider the constraints associated with this problem, which have been re-expressed as indicator functions in the objective. The set of permutation matrices, $\Pi_{|K_l|}$, is finite and thus it is clearly a semi-algebraic set. Notice that the set of weight matrices satisfying the norm bound is equivalent to $\{A : ||A||_2^2 - K_W < 0\}$. The function that defines this set is a polynomial, so it is a semi-algebraic set. Indicator functions on semi-algebraic sets are semi-algebraic functions. Thus, the indicator functions in the objective are semi-algebraic.

The graphs of semi-algebraic functions and subanalytic functions both belong to the logarithmic-exponential structure, an o-minimal structure. A basic algebraic property of o-minimal structures is that the graphs of addition and multiplication are also elements of the structure (van den Dries & Speissegger, 2002). Since our objective function is a linear combination of semi-algebraic functions and subanalytic functions, it follows that the graph of our objective function is an element of the logarithmic-exponential structure. Therefore, our objective function is a *tame* function and it satisfies the KL property.

### D.2 CONSIDERING RECTIFIED NETWORKS

Theorem 3.1 does not extend to the class of rectified networks. However, we are still interested in contructing a sequence of iterates $\{\phi^k, \boldsymbol{P}^k\}$ such that the objective value, $\mathbb{E}_{t \sim U}[\mathcal{L}(\boldsymbol{r}(t; \phi^k, \boldsymbol{P}^k))]$, is monotonic decreasing. The following theorem will introduce a technique for constructing such a sequence.

**Lemma D.1** ($\mathcal{L}$ restricted to possible network outputs is Lipschitz continuous). *For a feed-forward neural network, assume that $\mathcal{L}$ is continuous and that the neural network input, $X_0$, is bounded. Additionally, assume that the spectral norm of all weights, $\{W_i\}_{i=1}^L$, is bounded above by $K_W$, and the activation function, $\sigma$, is continuous with $||\sigma|| \leq 1$. Let $S_Y$ denote the set of $Y$ where*

$$Y = \boldsymbol{W}_L \sigma \circ \boldsymbol{W}_{L-1} \sigma \circ \boldsymbol{W}_{L-2}...\sigma \circ \boldsymbol{W}_1 \boldsymbol{X}_0 \tag{17}$$
$$such \; that \quad ||\boldsymbol{W}_i||_2 \leq K_W \quad \forall i \in \{1, 2, ..., L\}$$

*Then $\mathcal{L}$ restricted to the set $S_Y$ is Lipschitz continuous with some Lipschitz constant $K$.*

*Proof.* Since $X_0$ is bounded, it follows that there exists some constant $K_X$ such that $||X_0|| \leq K_X$. Since, the spectral norm of $W_1$ is bounded above by $K_W$, it is easy to see that $||W_1 X_0|| \leq K_W K_X$. Now since the pointwise activation function is a non-expansive map, it immediately follows that $||\sigma \circ W_1 X_0|| \leq K_W K_X$. Following this process inductively, we see that the network output, $Y$, is bounded and that:

$$||Y|| \leq K_W^L K_X \tag{18}$$

Since $Y$ is arbitrary, it follows that this is a bound for $S_Y$. Then we can restrict $\mathcal{L}$ to the ball in $\mathbb{R}^{m_L \times d}$ of radius $K_W^L K_X$. This ball is compact and $\mathcal{L}$ is continuous, so it follows that $\mathcal{L}$ restricted to this ball is Lipschitz continuous. Thus, there exists some Lipschitz constant $K$. Clearly, $S_Y$ is contained in this ball. Therefore, $\mathcal{L}$ is Lipschitz continuous on the set of all possible network outputs with Lipschitz constant $K$. $\qquad\square$

Let $\theta_1$ and $\theta_2$ be feed-forward neural networks with ReLU activation function. Assume that $\mathcal{L}$ and $r_\phi$ are piece-wise analytic functions in $C^1$ and locally Lipschitz differentiable. Assume that $\mathcal{R}$ is piece-wise analytic, lower semi-continuous, and bounded below. Assume that the maximum network width at any layer is $M$ units. Additionally, assume that the network weights have a spectral norm bounded above by $K_W$, and that this is a hard constraint when training the networks. Finally, any point on $r_\phi$ must be equivalent to an affine combination of neural networks (Bezier curves, polygonal chains, etc.) satisfying the previously stated spectral norm bound.

Create the parameterized curve $\boldsymbol{r}_\delta(t; \phi, \boldsymbol{P})$ by substituting the huberized ReLU function, $\sigma_\delta$, into all ReLU functions in $\boldsymbol{r}(t; \phi, \boldsymbol{P})$. We refer to the objective values associated with these curves as $Q_\delta(\phi, \boldsymbol{P})$ and $Q(\phi, \boldsymbol{P})$ respectively.

**Theorem D.2** (Monotonic Decreasing Sequence for Rectified Networks). *For a feed-forward network, assume the above assumptions have been met. Additionally, assume that $X_0$ is bounded, so that $\mathcal{L}$ restricted to the set of possible network outputs is Lipschitz continuous with Lipschitz constant $K_L$ by Lemma D.1. Now generate the sequence $\{\phi^k, \boldsymbol{P}^k\}$ by solving equation 6 for $r_\delta(t; \phi, \boldsymbol{P})$. On this sequence impose the additional stopping criteria that*

$$\frac{1}{2\nu_\phi}||\phi^{k+1} - \phi^k||_2^2 + \frac{1}{2\nu_P}||\boldsymbol{P}^{k+1} - \boldsymbol{P}^k||_2^2 \geq K_L\sqrt{M}\frac{\delta}{2}\sum_{i=1}^{L-1} K_W^i \qquad \forall k \geq 0. \qquad (19)$$

*Then, the sequence of curves $\boldsymbol{r}(t; \phi^k, \boldsymbol{P}^k)$ connecting rectified networks has monotonic decreasing objective value.*

*Proof.* First we consider the approximation error from replacing $\sigma$ with $\sigma_\delta$. It is straightforward to see that

$$\max_t |\sigma(t) - \sigma_\delta(t)| \leq \frac{\delta}{2}. \qquad (20)$$

Then it follows that for any input $\boldsymbol{x}$,

$$||\sigma \circ W_1\boldsymbol{x} - \sigma_\delta \circ W_1\boldsymbol{x}||_2 \leq \sqrt{M}\frac{\delta}{2}.$$

Since the spectral norm of $W_i$ are bounded above by $K_W$, then we see that

$$||W_2\sigma \circ W_1\boldsymbol{x} - W_2\sigma_\delta \circ W_1\boldsymbol{x}||_2 \leq K_W\sqrt{M}\frac{\delta}{2}.$$

Now notice that

$$||\sigma \circ W_2\sigma \circ W_1\boldsymbol{x} - \sigma_\delta \circ W_2\sigma_\delta \circ W_1\boldsymbol{x}|| \leq ||\sigma \circ W_2\sigma \circ W_1\boldsymbol{x} - \sigma \circ W_2\sigma_\delta \circ W_1\boldsymbol{x}|| \qquad (21)$$
$$+ ||\sigma \circ W_2\sigma_\delta \circ W_1\boldsymbol{x} - \sigma_\delta \circ W_2\sigma_\delta \circ W_1\boldsymbol{x}||.$$

Since the ReLU function is a non-expansive map, it must be that the first term is bounded above by the previous error, $K_W\sqrt{M}\frac{\delta}{2}$. The second term corresponds once again to the error associated with the huberized form of the ReLU function, $\sqrt{M}\frac{\delta}{2}$. Thus the total error can be bounded by $(K_W + 1)\sqrt{M}\frac{\delta}{2}$.

Following this inductively, it can be seen that the this error grows geometrically with the number of layers. Additionally, the loss function is Lipschitz continuous when restricted to the set of possible network outputs. So we find the following bounds:

$$||Y - Y_\delta|| \leq \sqrt{M}\frac{\delta}{2}\sum_{i=1}^{L-1} K_W^i$$

$$||\mathcal{L}(Y) - \mathcal{L}(Y_\delta)|| \leq K_L\sqrt{M}\frac{\delta}{2}\sum_{i=1}^{L-1} K_W^i \qquad (22)$$

Since any point on the curve is an affine combination of networks with the $K_W$ bound on the spectral norm of their weights, it immediately follows this spectral norm bound also holds for the weights for any point on the curve. Then $||Q(\phi, \boldsymbol{P}) - Q_\delta(\phi, \boldsymbol{P})||$ is also bounded above by the bound in equation 22.

Then let $\{\phi^k, \boldsymbol{P}^k\}$ be the sequence generated by solving equation 6 using the curve $r_\delta$. $\sigma_\delta$ is a piecewise analytic function in $C^1$ and is locally Lipschitz differentiable. Additionally, the spectral norm constraint on the weights is semi-algebraic and bounded below, so Theorem 3.1 can be applied. It then follows that

$$Q(\phi^{k+1}, \boldsymbol{P}^{k+1}) + \mathcal{R}(\phi^{k+1}) + \frac{1}{2\nu_\phi}||\phi^{k+1} - \phi^k||_2^2 + \frac{1}{2\nu_P}||\boldsymbol{P}^{k+1} - \boldsymbol{P}^k||_2^2 \qquad (23)$$

$$\leq Q(\phi^k, \boldsymbol{P}^k) + \mathcal{R}(\phi^k) + K_L\sqrt{M}\frac{\delta}{2}\sum_{i=1}^{L-1} K_W^i, \qquad \forall k \geq 0$$

Thus, $\boldsymbol{r}(t; \phi^k, \boldsymbol{P}^k)$ is a sequence of curves, connecting rectified networks, with monotonic decreasing objective value as long as

$$\frac{1}{2\nu_\phi}||\phi^{k+1} - \phi^k||_2^2 + \frac{1}{2\nu_P}||\boldsymbol{P}^{k+1} - \boldsymbol{P}^k||_2^2 \geq K_L \sqrt{M} \frac{\delta}{2} \sum_{i=1}^{L-1} K_W^i \qquad \forall k \geq 0$$

Since the above equation is a stopping criterion introduced in the theorem statement, it follows that we have constructed a sequence of curves, connecting rectified networks, with monotonic decreasing objective value. $\qquad \square$

## E  RESIDUAL NETWORK ALIGNMENT

Algorithm 1 applies to networks with a typical feed-forward structure. In this section, we discuss how we compute alignments for the ResNet32 architecture as it is more complicated. It is important to align networks such that the network structure is preserved and network activations are not altered. In the context of residual networks, special consideration must be given to skip connections.

Consider the formulation of a basic skip connection,

$$\boldsymbol{X}_{k+1} = \sigma \circ (\boldsymbol{W}_{k+1} \boldsymbol{X}_k) + \boldsymbol{X}_{k-1} \tag{24}$$

In this equation, we can see that $\boldsymbol{X}_{k+1}$ and $\boldsymbol{X}_{k-1}$ share the same indexing of their units. This becomes clear when you consider permuting the hidden units in $\boldsymbol{X}_{k-1}$ without permuting the hidden units of $\boldsymbol{X}_{k+1}$. It is impossible to do so without breaking the structure of the equation above, where there is essentially the use of an identity mapping from $\boldsymbol{X}_{k-1}$ to $\boldsymbol{X}_{k+1}$.

We consider a traditional residual network that is decomposed into residual blocks. In each block the even layers have skip connections while the odd layers do not. So, we compute the alignment as usual for odd layers. For all even layers within a given residual block, we determine a shared alignment. We do this by solving the assignment problem for the average of the cross-correlation matrix over the even layers in that residual block.

## F  ALIGNMENT ALONG CURVES

Clearly, alignment is a useful method for learning better flat loss curves between models. An interesting question is how curve finding itself relates to alignment. Until now, we have only considered the alignment between the endpoint models, $\boldsymbol{r}(0)$ and $\boldsymbol{r}(1)$. Now, we consider how points along the curve, $\boldsymbol{r}(t)$, align to the endpoints. To study this numerically, we will use the curve midpoint $\boldsymbol{r}(0.5)$. From Figure 4, we see that this is the point on the quadratic Bezier curve that is roughly linearly connected to both endpoints.

### F.1  CORRELATION SIGNATURE

First, we consider how the correlation signature changes along the curve. Figure 14 displays the correlation signature between the curve midpoint and each endpoint in blue. To gain a better understanding of this signature, we require some context. Thus, the correlation signature between the linear midpoint and each endpoint is displayed in green. This allows us to understand how the correlation signature changes over the curve finding optimization. Additionally, we display the correlation signature between the curve midpoint and each endpoint, where the midpoint has been aligned to the given endpoint, in yellow. This essentially gives us context on how highly the midpoint is aligned to each endpoint. This is because the yellow curve acts as an upper bound for the blue curve.

There are several observations to be made about Figure 14. The correlation signature between the endpoint and the curve midpoint is fairly high. For unaligned endpoints, the correlation is only slightly lower than the signature computed when the curve midpoint is aligned to the endpoint. In the case where the endpoints are aligned, the signatures are seen to coincide. This suggests that the curve finding algorithm is finding the quadratic curve along which similar feature representations are being interpolated.

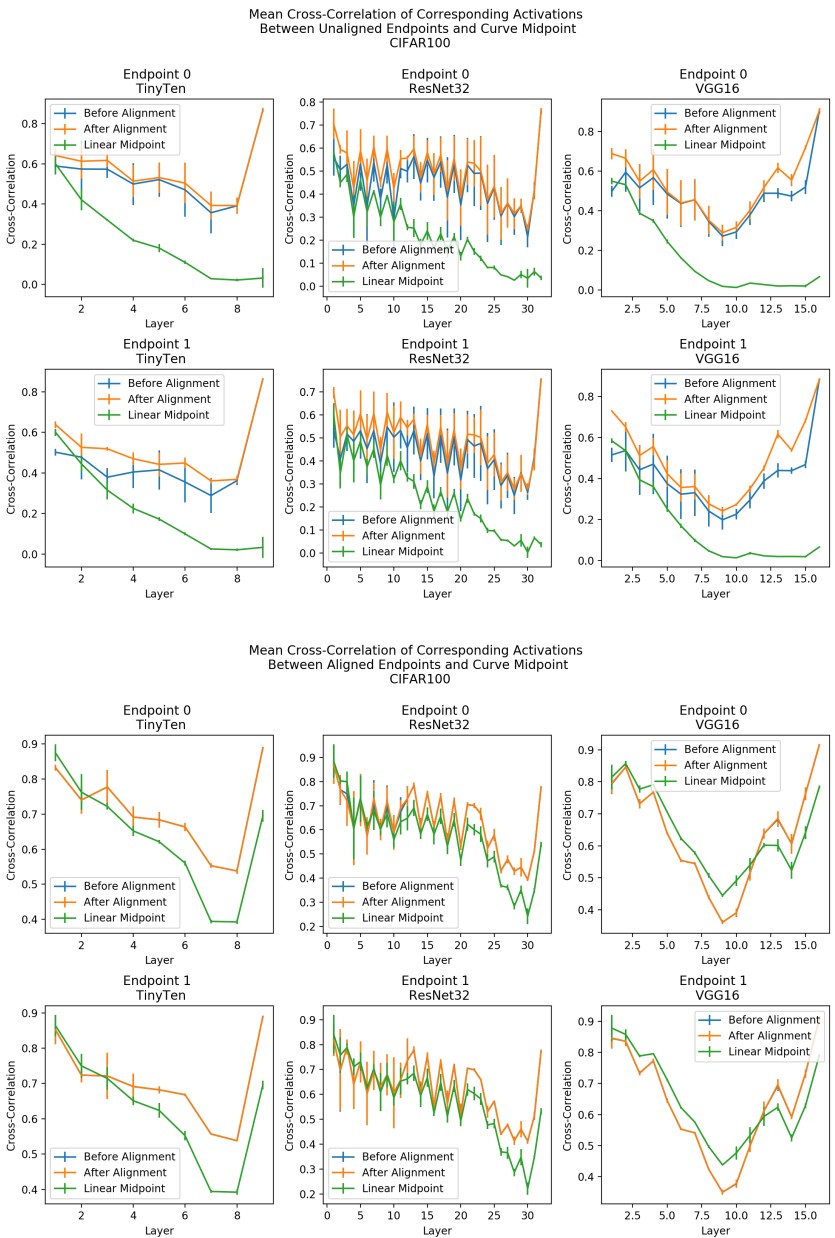

Figure 14: The mean cross-correlation between units in the curve midpoint model and each endpoint model. For context, the mean cross-correlation between the linear midpoint and each endpoint is displayed. Additionally, the mean cross-correlation between the curve midpoint and each endpoint after being aligned to the respective endpoint is displayed.

Concerning the linear midpoint, the correlation at the linear midpoint decays to 0 when endpoints are unaligned as the network goes deeper. When endpoints are aligned, the correlation signature at the linear midpoint is similar to the correlation signature at the curve midpoint. Since these linear connections between the endpoints are the initializations for the curve finding algorithm, this gives some intuition on how alignment works to give a better initialization.

