# OpenReview forum: "Optimizing Loss Landscape Connectivity via Neuron Alignment"
_ICLR.cc/2020/Conference — Reject_

### Official Review · AnonReviewer1 · 2019-10-22
**Official Blind Review #1**

**Rating:** 1

**Review:**

Given two parameters theta_1 and theta_2 of the same architecture, the authors propose to learn a minimal loss curve between theta_1 and P theta_2, where P is a permutation matrix yielding another equivalent parameterization of the same network. The authors show that either by initializing P with neuron alignment or by optimizing P jointly with the curve, one can find a parameter-space with better accuracy and loss than by just learning a curve between theta_1 and theta_2. The authors also show that initializing P is sufficient to obtain these gains, avoiding the complexity of also optimizing for P. Furthermore, they show that ensembles across models from these curves have a very mild gain in accuracy to those of non-aligned curves.

The main qualm I have about this paper is about the significance of the contributions and the motivation.
At the core, the authors propose to find a curve between theta_1 and P theta_2 where P comes from aligning theta_1 and theta_2 as in (Li et al. 2016.). This is lacking almost any motivation, or discussion on what problem they are trying to solve. Are they trying to find ensembles with lower error? If that is the case, well the results are evidence of a negative result in this respect, which is okay but given how ad-hoc and poorly motivated the method is to that objective it's not much of a contribution. Are they trying to better understand theoretically the loss landscape of neural networks? I don't think there's any theoretical gain in that regard from this paper either. They show that the curves between aligned networks are better, but they don't show how this relates to anything else in the published literature or open questions in the theoretical deep learning field.

Regarding contribution 2, PAM is in the end shown to not converge to better curves than simply initializing with alignment. Also, doesn't the convergence result to a critical point also apply for standard descent methods? The convergence theorem doesn't seem to be much of a contribution in my opinion, either to the optimization or the deep learning community.

Regarding contribution 3, I agree that better curves can be learned faster, but why is this a meaningful contribution? What problem that the deep learning or the theoretical machine learning cares about does this help solve?

Regarding contribution 4, as the authors themselves admit, the improvement is very dim in comparison to non-aligned curves, and comparisons to other ensemble methods are not present.

I want to acknowledge that while the motivation and contributions are in my opinion dim, I find the experimental methodology of this paper very solid. All the claims are to my extent very well validated with experiments, and the experiments are in general well designed to validate those claims. My problem is sadly not with the validity of the claims but with their significance.

**Experience Assessment:**

I have read many papers in this area.

**Review Assessment: Checking Correctness Of Derivations And Theory:**

I assessed the sensibility of the derivations and theory.

**Review Assessment: Checking Correctness Of Experiments:**

I assessed the sensibility of the experiments.

**Review Assessment: Thoroughness In Paper Reading:**

I read the paper at least twice and used my best judgement in assessing the paper.

---

> ### Author Response · Authors · 2019-11-15
> **Thank you for your comments**
>
> We would like to thank the reviewer for their feedback and their time.
>
> Regarding the main qualms, please see our general comment on the motivation and contributions of this work.
>
> Regarding contribution 2, please see our general comment discussing PAM.
>
> Regarding contribution 3, we are interested in understanding the loss landscape of trained neural networks by exploring mode connectivity.  In the context of this paper, training the curve can be seen as training an ensemble for a very low cost.  Our results show that the mode connectivity via neuron alignment provides a method to identify models on the path that have low training loss as well as high accuracy and show a modest improvement in accuracy upon ensembling. As mode connectivity is an active research topic, we believe others could find this faster and efficient path training useful if their work does not require fixed symmetry.
>
> Regarding contribution 4, please see our general comment on the significance of the ensembling results.
>
> Again, we thank the reviewer for their time and comments. Additionally, we appreciate the comments on the experimental methodology of this paper. We hope that we have resolved much of the misunderstanding regarding the motivation and contributions of this work.

---

### Official Review · AnonReviewer3 · 2019-10-23
**Official Blind Review #3**

**Rating:** 6

**Review:**

The paper investigates the connection between symmetries in the neural network architectures and the loss landscape of the corresponding neural networks. In the previous works, there was shown that the two local minima of a neural network can be connected by a curve with the low validation/train loss along the curve. Despite the loss on the curve being close to the loss at the end points, there are segments of the curve on which the loss in higher than loss at the local minima. To overcome this problem, the authors proposed two-step procedure: 1. Align weights between two neural networks 2. Launch the path finding algorithm between aligned weights. In other words, the authors proposed to connect not original local minima but local minima that parametrize the same functions as the original ones, but have a different order of neurons. The authors also proposed PAM algorithm where they iteratively apply path finding algorithm and weights alignment algorithm.

Novelty and significance. The idea to combine the symmetrization of NN architectures with path finding algorithms is new to the best of my knowledge. Experimentally, the authors showed that ensembling the midpoint and endpoints of the curve found via path finding  algorithm coupled with the neural alignment algorithm delivers a better result than simple averaging of three independent networks. This is a new and significant result, since before the ensembling of points along the curve had the same quality as the ensemble of three independent networks or marginally better.
The weak side of the paper is the PAM algorithm that occupies a significant part of the paper and does not deliver a significantly better result than the simple application of the neural alignment procedure before launching the path finding algorithm.

Clarity. Overall, the related work section contains all relevant references to the previous works to the best of my knowledge. The paper is well written, excluding the section Neuron Alignment that lacks notation description.

The paper contains several typos and inaccuracies:
1. “However, We find its empirical performance is similar to standard curve finding, and we advocate the latter for practical use due to computational efficiency.” The word “We” should start with the lowercase letter.
2. In the sentence “The first question to address is how to effectively deal with the constraints in equation 6” the index i should be replaced with the index l.
3. Notation Π|Kl| introduced after equation 5.
4. In the section describing neuron alignment  algorithm Lie et al. [1] used a different notation. So I would recommend to further extend this section and add all necessary notations. Also, I would recommend to add a direct link to the paper where the problem is described in matrix form.
5. In the Algorithm 1 “Initialize P θ2 := [Wˆ 2 1 ,Wˆ 2 2 , ...,Wˆ 2 L ] as [W2 1 ,W2 2 , ...,W2 L ] for k ∈ {1, 2, ..., L − 1};”  k is not used anywhere in notation.
6. In Figure 3, “model 2 aligned ” sing is out of the plot box for ResNet-32 and VGG-16 architectures.
7. The appendix contains the sketch of the proof that is quite difficult to follow. I would recommend giving all necessary definitions as it is done in the [2] and extend the proof.

Overall, it is quite an interesting paper but it contains some drawbacks.
[1] Yixuan Li, Jason Yosinski, Jeff Clune, Hod Lipson, and John E Hopcroft.  Convergent learning: Do different neural networks learn the same representations?  In ICLR, 2016

[2]  H́edy Attouch, J́erˆome Bolte, Patrick Redont, and Antoine Soubeyran.  Proximal alternating minimization and projection methods for nonconvex problems:  An approach based on the kurdyka-łojasiewicz inequality.Mathematics of Operations Research, 35(2):438–457, 2010


**Experience Assessment:**

I have published one or two papers in this area.

**Review Assessment: Checking Correctness Of Derivations And Theory:**

I assessed the sensibility of the derivations and theory.

**Review Assessment: Checking Correctness Of Experiments:**

I carefully checked the experiments.

**Review Assessment: Thoroughness In Paper Reading:**

I read the paper at least twice and used my best judgement in assessing the paper.

---

> ### Author Response · Authors · 2019-11-15
> **Thank you for your review**
>
> We thank the reviewer for their feedback and their time. We appreciate the reviewer sharing typos and inaccuracies with us. In our revised paper, we have corrected for those we immediately agree with. For others, we address them below.
>
> PAM: Please see our general comment discussing the role of PAM in this paper.
>
> 4. We have revised the section on the background for Neuron Alignment. We have focused on increasing readability and providing references for the linear assignment problem. Our notation is consistent with Li et al. (2016) with some alterations made for the purpose of clarity.
>
> 7. We have provided more detail in the proof in Section D.1. This mainly includes more definitions related to showing that the objective function satisfies the Kurdyka-Lojawiesics property. We believe that the proof should be much easier to follow now.
>
> Once again, we would like to thank the reviewer for their time.

---

### Official Review · AnonReviewer2 · 2019-10-26
**Official Blind Review #2**

**Rating:** 3

**Review:**

This paper combines neuron alignment with mode connectivity. Specifically, it applies paths neuron alignment to the calculation of mode-connecting and empirical results show that alignment helps in finding better curves.
Combining neuron alignment with mode connectivity seems to be a good idea, but the message the authors want to convey is somewhat vague. Some key details are not presented clearly, and some contents seem to be irrelevant. The following are some detailed comments and questions:
1. One main contribution of this paper is the observation that the observation that alignment helps in finding better curves. An observation is excellent if it brings significant performance improvements in practice, or if it brings deep insights in the understanding of the field. However, for the former, the improvement in the performance is not that much; for the latter, there is hardly any insight conveyed by this paper. Therefore, this observation itself is not strong enough.
2. One contribution of this paper is applying proximal alternating minimization (PAM) when optimizing the parameters and proving its convergence. Nonetheless, PAM is only used in one model (TinyTen) and does not bring any improvement in the performance. It seems that there is no point in applying PAM and the contents related to PAM are all somewhat irrelevant.
3. Usually sufficient details help in good understandings, but in this paper, some key details are unfortunately missing. For example, in Algorithm 2, details on the optimization step is not clear: what is the optimization method the authors use other than PAM? Also, no comments are addressed on Figure 7 to Figure 10, either in the main body or in the appendix. I would like to see more explanations details. If the source code is provided, it will be better.
In sum, the idea seems to be interesting, but the overall quality of the paper is still yet to be improved, and some key details need to be addressed more clearly before it can be accepted as a qualified submission.


**Experience Assessment:**

I have published one or two papers in this area.

**Review Assessment: Checking Correctness Of Derivations And Theory:**

I assessed the sensibility of the derivations and theory.

**Review Assessment: Checking Correctness Of Experiments:**

I assessed the sensibility of the experiments.

**Review Assessment: Thoroughness In Paper Reading:**

I read the paper at least twice and used my best judgement in assessing the paper.

---

> ### Author Response · Authors · 2019-11-15
> **Thank you for your review**
>
> We would like to thank the reviewer for their feedback and their time.
>
> 1. Please see our general comment on the motivation and contributions of this work. The other main empirical result of this paper concerned ensembling performance. Please see our general comment addressing our ensembling results.
>
> 2. Please see our general comment discussing the role of PAM in this paper.
>
> 3. We apologize for lack of detail in places. We now provide additional detail in the Experiment section on the methodology for learning the curves. We have fixed the lack of references to figures in the appendix. Also, we have anonymously shared the code with all reviewers and the AC in a private comment. Once again, we thank the reviewer for their time.

---

### Author Response · Authors · 2019-11-15
**Motivation and Contributions**

We address the motivation of this work in the second and third paragraphs of  Introduction. The study of finding optimal curves between models, also known as mode connectivity, has been of recent interest in the deep learning community (Freeman & Bruna (2016); Garipov et al. (2018); Gotmare et al. (2018)). The parameterization of the models may still contain a weight ambiguity, i.e., the neurons in the same positions of different models but same architecture may not correspond to each other. Consequently, the curves connecting the models could fail to interpolate similar feature representations, caused by these so-called barriers, which in turn could break a critical structure in the interpolated networks. To this end, we want to know to what extent barriers between neural networks along optimal curves are truly just artifacts of symmetry. Understanding this problem could provide insight into the dynamics of training a neural network. Then explicitly, we are interested in finding low loss curves between networks up to symmetry, to better understand the loss landscape.

We reiterate the contributions summarized in Introduction. We formally generalize the curve  finding problem to account for permutation ambiguity. We introduce a rigorous framework for solving this problem theoretically, known as proximal alternating minimization (Attouch (2010)). Establishing convergence for a nice subset of networks is a part of what makes it rigorous. We introduce neuron alignment as an inexpensive heuristic for approximating this permutation. Empirically, we show that the consideration of this alignment is critical for learning nearly flat loss curves which generalize better. PAM allows us to see that the permutation from neuron alignment is near the local optimal permutation. Lastly, we see a modest to notable improvement to ensembling, that is particularly notable for underparameterized models. We have added results from the new and updated experiments as well as additional text to clarify the significance of our results. Additionally, we now reference Appendix F in the main body of the text. This appendix gives insight into how the curve finding algorithm is related to neuron alignment.

---

### Author Response · Authors · 2019-11-15
**Addressing Ensembling Performance**

The results suggest that the performance increase upon ensembling  is significant for simple architectures, which correspond to TinyTen in our original experiments. To further highlight the strength of ensembling  on the aligned path  for underparameterized networks, we introduce the TinySix architecture to the paper. It is a a modified version of TinyTen with 4 layers removed. The results are consistent and in line  with earlier literature (Ju et al. 2018),  reporting more evident performance gain from ensembling for simpler architectures. It is noteworthy that the averaging technique used in the present study is the same as used in (Garipov et al. (2018)). We do forgo analyzing more complicated ensemble methods, which is a common practice even for papers with a main focus on ensembling. While the improvement may seem marginal as some of the reviewers have stated, it should be noted that the magnitude of the accuracy increase  is similar to what was reported  in (Garipov et al. (2018)), when their Fast Geometric Ensemble method was compared to the earlier Snapshot ensembling in (Huang et al. (2017)). This observation highlights the significance of neuron alignment for better ensembling, which is now made clear in the paper.

---

### Author Response · Authors · 2019-11-15
**On the Role of PAM**

First, we apologize that our reasoning for including PAM into this paper was not clear to the reviewers. To this end, we have revised the paper to eliminate the perceived disconnect of PAM from the rest of the paper. We have also worked to make the focus of this work more concise to the reviewer. Listening to the reviewers, we have made non-minor revisions to this work. To aid reviewers, we have highlighted key differences/additions in the new text.

We also include updated results in the revised version of this paper. The objective function used in PAM was modified so that the control point of the Bezier curve is a function of the permutation $P$ (See Equation 8 of the revised paper). The updated results can be found in the revised Table 2. We have introduced PAM results for the ResNet32 architecture, and a new architecture introduced in the revised paper called TinySix. We note in the paper that it was too computationally intensive to train curves using PAM for VGG16. We find that the PAM Unaligned case now outperforms the Unaligned case. In this new implementation, PAM displays a performance benefit in its own right.

Overall, the reason for including PAM in this work is to provide a a theoretical framework for addressing the generalized optimization problem in Equation 5 through a rigorous method. Part of what makes this approach rigorous is our ability to establish convergence guarantees for a subset of "nice enough" neural networks. We note that establishing convergence for alternating minimization methods is typically nontrivial. The PAM Aligned case gives us an upper bound for judging the performance of the Aligned case. Since the performance of these two methods is comparable, this suggests that the aligned permutation $P_{Al}$ is already close to a locally optimal permutation. This is ideal as the aligned curves are much less expensive to compute than the PAM Aligned curves. Then we have an approximation method to solve curve finding up to symmetry for large cases like VGG16 where PAM becomes computationally infeasible. We strongly emphasize this motivation in the revised paper.

---

### Decision · Program_Chairs · 2019-12-19

**Decision:**

Reject

**Comment:**

This paper studies the loss landscape of neural networks by taking into consideration the symmetries arising from the parametrisation. Specifically, given two models $\theta_1$, $\theta_2$, it attempts to connect $\theta_1$ with the equivalence of class of $\theta_2$ generated by weight permutations.
Reviewers found several strengths in this work, from its intuitive and simple idea to the quality of the experimental setup. However, they also found important shortcomings in the current manuscript, chief among them the lack of significance of the results. As a result, this paper unfortunately cannot be accepted in its current form. The chairs encourage the authors to revise their work by taking the reviewer feedback into consideration.